# Responses of polar energy budget to regional sea surface temperature changes in extra-polar regions

**Qingmin Wang, Yincheng Liu, Lujun Zhang, and Chen Zhou**

School of Atmospheric Sciences, Nanjing University, Nanjing, 210023, China

**Correspondence:** Chen Zhou (czhou17@nju.edu.cn)

**Abstract.** Surface temperature at polar regions not only is affected by local forcings and feedbacks but also depends on teleconnections between polar regions and low-latitude regions. In this study, the responses of the energy budget in polar regions to remote sea surface temperature (SST) changes are analyzed using a set of idealized regional SST perturbation experiments. The results show that the responses of the polar energy budget to remote sea surface warmings are regulated by changes in atmospheric energy transport, and radiative feedbacks also contribute to the polar energy budget at both the top of atmosphere (TOA) and the surface. Specifically, an increase in poleward atmospheric energy transport to polar regions results in an increase in surface and air temperature, and the corresponding Planck feedback leads to radiative warming at the surface and radiative cooling at the TOA. In response to sea surface warmings in most mid-latitude regions, poleward atmospheric energy transport to polar regions in the corresponding hemisphere increases. Sea surface warming over most tropical regions enhances the polar energy transport to both Arctic and Antarctic regions, except that an increase in the Indian Ocean's temperature results in a decrease in poleward atmospheric energy transport to the Arctic due to the different responses of stationary waves. The sensitivity of the Arctic energy budget to tropical SST changes is generally stronger than that of the Antarctic energy budget, and poleward atmospheric heat transport is dominated by dry static energy, with a lesser contribution from latent heat transport. The polar energy budget is not sensitive to SST changes in most subtropical regions. These results help to explain how the polar climate is affected by the magnitude and spatial pattern of remote SST change.

## 1 Introduction

As the global surface temperature increases, the Arctic region has experienced a surface temperature rise of more than twice the global average (Lenssen et al., 2019), a phenomenon known as "Arctic amplification" (AA). On the other hand, Antarctic warming is weaker compared to Arctic warming due to the higher average elevation of the Antarctic continent, lower albedo, feedback efficiency differences, and the Southern Ocean's heat absorption capacity (Marshall, 2003; Salzmann, 2017; Smith et al., 2019; Hahn et al., 2021). The polar energy budget is highly sensitive to various local feedback mechanisms. One important mechanism is the ice–albedo feedback. Global warming reduces snow cover and sea ice cover in the polar regions, leading to more solar radiation being absorbed, which in turn accelerates climate warming and further decreases albedo (Dickinson et al., 1987; Hall, 2004; Boeke and Taylor, 2018; Duan et al., 2019; Dai et al., 2019). Temperature feedback is another significant contributor to AA (Pithan and Mauritsen, 2014; Laîné et al., 2016; Sejas and Cai, 2016). It involves the processes of radiative cooling and is characterized by the Planck and lapse-rate feedbacks. The Planck feedback, driven by the nonlinear relationship between blackbody radiation and temperature, provides negative feedback to top of atmosphere (TOA) fluxes at all latitudes, especially in low latitudes (Pierrehumbert, 2010). The lapse-rate feedback is a significant driver of AA: in the Arctic regions, stable stratification and temperature inversions trap surface warming and reduce radiative cooling, thereby enhancing warming. In contrast, the tropics experience significant upper-atmosphere warming due to convection, which

does not similarly trap heat (Pithan and Mauritsen, 2014). During climate warming, the transformation of ice clouds into water clouds increases cloud albedo, leading to negative feedback (Mitchell et al., 1989; Li and Le Treut, 1992). Simultaneously, the decrease in lower-tropospheric stability increases Arctic cloud cover and optical thickness (Barton et al., 2012; Solomon et al., 2014; Taylor et al., 2015; Yu et al., 2019), contributing to Arctic autumn and winter warming (Boeke and Taylor, 2018). These local feedbacks are considered the primary contributors to AA (Pithan and Mauritsen, 2014; Goosse et al., 2018; Hahn et al., 2021; Dai et al., 2019).

The polar climate is also affected by remote influences, whose interaction drives Arctic warming (Li et al., 2021). While some studies suggest that remote forcing plays a relatively minor role in AA (Stuecker et al., 2018), other research highlights the significant impact of poleward heat and moisture transport from lower latitudes in enhancing Arctic warming, and AA exists even in the absence of local sea ice feedbacks (Alexeev et al., 2005; Graversen and Burtu, 2016). Specifically, poleward atmospheric heat transport (AHT) and moisture transport are critical components that contribute substantially to the observed warming in the Arctic.

Under global warming, the AHT from low latitudes is more effective in reaching the polar regions compared to the equatorward transfer from high latitudes (Alexeev et al., 2005; Chung and Räisänen, 2011; Park et al., 2018; Shaw and Tan, 2018; Semmler et al., 2020), and multiple global climate model experiments have been conducted to measure the remote influence on Arctic warming (Alexeev et al., 2005; Chung and Räisänen, 2011; Yoshimori et al., 2017; Park et al., 2018; Shaw and Tan, 2018; Stuecker et al., 2018; Semmler et al., 2020). The transport of water vapor from mid-latitudes also plays an important role by enhancing the greenhouse effect prior to condensation and increasing cloudiness after condensation, which together warm the Arctic during winter (Graversen and Burtu, 2016). Graversen and Burtu (2016) showed that latent heat transport can lead to significantly more Arctic warming than dry static energy (DSE) transport, even when delivering an equivalent amount of energy. Therefore, remote processes play an important role in driving Arctic warming, and the remote forcings are further amplified by local feedback processes.

In low-latitude regions, sea surface temperature (SST) variations markedly affect the polar energy budget (Alexeev et al., 2005). It is widely accepted that planetary waves play a critical role in establishing teleconnections between tropical oceans and polar regions. These waves are pivotal in the transport of heat and moisture to the Arctic, consequently driving the increase in polar temperatures (Graversen and Burtu, 2016; Baggett and Lee, 2017). For instance, intensified convective activity within the Pacific Warm Pool not only strengthens the propagation of Rossby waves toward the poles but also increases the frequency of these fluctuations. This enhancement in Rossby wave activity boosts the transport of water vapor to the Arctic, augmenting the downward

longwave radiation in the Arctic regions (Rodgers, 2003; Lee et al., 2011; Lee, 2012, 2014). While synoptic-scale transient eddies contribute significantly to mean-state poleward heat transport and its changes under increased $CO_2$ (Donohoe et al., 2020), their overall impact is relatively minor compared to that of amplified planetary waves in response to tropical warming (Baggett and Lee, 2017). Atmospheric circulation models reveal that the warming of tropical SST from glacial to interglacial periods significantly elevates summer temperatures in regions where the Canadian ice sheet (also known as the Laurentide ice sheet) forms. Conversely, cold tropical SST perturbations exert a lesser impact on water vapor transport and temperature in the Canadian region (Rodgers, 2003). The tropical SST anomalies during El Niño and La Niña events have a large impact on Arctic surface air temperature (Lee et al., 2011; Lee, 2012, 2014), but the impacts of major El Niño events on Arctic temperatures are distinct due to differences in eastern tropical Pacific SST (Jeong et al., 2022), indicating that the amplitude and spatial pattern of SST change are important for Arctic climate predictions.

In previous research, scholars primarily focused on the impact of SST on the polar energy budget over a large area of tropical oceans. However, studies have indicated significant variations in the effects of SST anomalies on the global climate system in different oceanic regions (Barsugli and Sardeshmukh, 2002; Fletcher and Kushner, 2011). In this study, we employ a set of idealized SST patch experiments to perform a systematic analysis of the response of the polar energy budget to SST changes in various areas.

## 2   Data and method

### 2.1   Individual patch experiments

The patch experiments were conducted using the Community Earth System Model version 1.2.1 integrated with the Community Atmospheric Model version 5.3 (CESM1.2.1-CAM5.3; Neale et al., 2012), operating at a spatial resolution of 1.9° latitude by 2.5° longitude. The experimental setup included a control experiment and two sets of patch experiments – one with warm patches and another with cold patches. The control experiment spanned 41 years, maintaining the SST, sea ice, and climatic forcings at the constant present-day levels observed (in the year 2000). The global ocean was segmented into 80 overlapping rectangular areas, comprehensively covering the global ice-free ocean surface, as depicted in Fig. 1 of Zhou et al. (2020). In the warm patch experiments, a positive SST anomaly was introduced at the ocean surface within a designated patch, while the SST in other regions was the same as the control setup. Conversely, the cold patch experiments involved introducing a negative SST anomaly at the ocean surface within the respective patches. The SST anomalies in each patch were designed according to the equation proposed by Barsugli and Sardeshmukh (2002), which effectively mitigates nonlinearity due to

unrealistic SST gradients, ensuring a more realistic simulation of oceanic temperature variations:

$$\Delta \text{SST}_P(\text{lat}, \text{long}) =$$
$$A \cos^2 \left( \frac{\pi}{2} \frac{\text{lat} - \text{lat}_p}{\text{lat}_w} \right) \left( \frac{\pi}{2} \frac{\text{long} - \text{long}_p}{\text{long}_w} \right), \quad (1)$$

where $|\text{lat} - \text{lat}_p| < \text{lat}_w$, $|\text{long} - \text{long}_p| < \text{long}_w$. The terms lat$_p$ and long$_p$ are the latitude and longitude of the center point for a specific patch, respectively; lat$_w$ and long$_w$ are the meridional and zonal half-width of the patch, respectively, with their values set to lat$_w = 10°$ and long$_w = 40°$ in these experiments; and $A$ is the amplitude of the SST anomaly, which is set to be $+4$ and $-4$ K in this study.

In this study, we analyzed the responses of polar TOA radiative fluxes ($R_{\text{TOA}}$), polar surface fluxes ($R_{\text{sfc}}$), and heat fluxes resulting from atmospheric heat transport to the polar regions ($R_{\text{AHT}}$). The equations for calculating these parameters are listed as follows:

$$R_{\text{TOA}} = \text{FSNT} - \text{FLNT}, \quad (2)$$
$$R_{\text{sfc}} = \text{FSNS} - \text{FLNS} - \text{SH} - \text{LH}, \quad (3)$$
$$R_{\text{AHT}} = R_{\text{sfc}} - R_{\text{TOA}}, \quad (4)$$

where FSNT represents the net downward shortwave radiation at the TOA; FLNT denotes the net upward longwave radiation at the TOA; FSNS is the net downward shortwave radiation at the surface; FLNS represents the net upward longwave radiation at the surface; and SH and LH account for the sensible and latent heat fluxes, respectively. Additionally, both SH and LH are defined as positive upward.

## 2.2 EOF-SST experiments

To quantify the polar energy budget response to realistic SST anomaly patterns, we applied an empirical orthogonal function (EOF) analysis to historical SST data from 1980 to 2019, obtained from the Hadley Centre Sea Ice and Sea Surface Temperature dataset (HadISST; Rayner et al., 2003), and identified the first eight EOF modes. The first eight EOF modes explain approximately 55 % of the total variance in global SST anomalies, thereby representing the predominant variability patterns. We then conducted eight separate EOF-SST experiments, and the SST of each experiment was perturbed by a specific EOF mode relative to the control run. These experiments allow us to isolate and understand the impact of realistic SST anomaly patterns on the polar energy budget.

## 2.3 Radiative kernel decomposition methodology

This study employs the radiative kernel approach (Soden et al., 2008; Huang et al., 2017) to decompose both surface and TOA radiation into the radiative effects of various meteorological variables, measured in watts per square meter (W m$^{-2}$). The core calculation involves multiplying the radiative kernels by the monthly anomalies of the corresponding climate fields as follows:

$$\Delta R_X = K_X \cdot \Delta X, \quad (5)$$

where $X$ denotes an arbitrary non-cloud climate variable; $\Delta R_X$ represents the radiative effect at the surface or TOA associated with that variable; $K_X$ is the corresponding radiative kernel; and $\Delta X$ is the monthly anomaly of the climate variable, calculated as the deviation from the monthly climatological average. Positive values of $\Delta R$ indicate an increase in net incoming radiation, which corresponds to a warming effect on the Earth. The radiative kernels used in this analysis are derived from the ERA-Interim climatological fields and have been validated to perform well with the climate model surface outputs (Huang et al., 2017; Liu et al., 2024).

Cloud radiative effects are calculated following the methodology of Soden et al. (2008):

$$\Delta R_{\text{cld}} = \Delta \text{CRF} - \sum_X \left( K_X - K_X^0 \right) \Delta X. \quad (6)$$

In this equation, $\Delta R_{\text{cld}}$ denotes the cloud-induced radiative anomalies, and CRF (cloud radiative forcing) is defined as the difference in surface net radiation fluxes between all-sky and clear-sky conditions. The superscript 0 means the clear-sky kernels.

Building upon this framework, the study further decomposes the TOA and surface radiative anomalies into specific feedback components to achieve a more detailed analysis of the factors influencing the Earth's radiation balance. $\Delta R_{\text{TOA}}$ is partitioned into cloud-induced radiative anomalies ($\Delta R_{\text{TOA,cld}}$), albedo-induced radiative anomalies ($\Delta R_{\text{TOA,alb}}$), Planck-feedback-induced radiative anomalies ($\Delta R_{\text{TOA,plk}}$), and lapse-rate-feedback-induced radiative anomalies ($\Delta R_{\text{TOA,LR}}$). Similarly, $\Delta R_{\text{sfc}}$ is broken down into cloud-induced surface radiative anomalies ($\Delta R_{\text{sfc,cld}}$), albedo-related surface radiative anomalies ($\Delta R_{\text{sfc,alb}}$), Planck-feedback-induced surface radiative anomalies ($\Delta R_{\text{sfc,plk}}$), lapse-rate-feedback-induced surface radiative anomalies ($\Delta R_{\text{sfc,LR}}$), LH anomalies ($\Delta \text{LH}$), and SH anomalies ($\Delta \text{SH}$).

# 3 Results

## 3.1 Responses of polar energy budget to local SST changes

The differences in the annual polar energy budget in conjugate SST warming patch experiments and SST cooling patch experiments are shown in Fig. 1. The location of each point denotes the center of the corresponding patch, and the colors of these points denote the differences in the polar energy budget between corresponding conjugate warming and cooling patch experiments. Additionally, $t$ tests were conducted to assess their statistical significance. The difference between

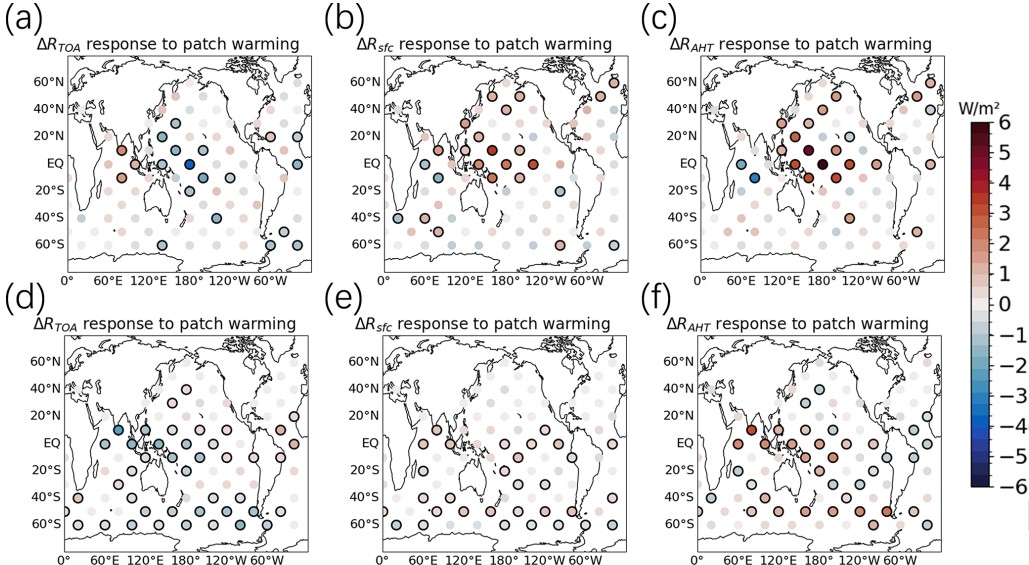

**Figure 1.** Responses of the polar energy budget to regional SST changes. **(a)** Differences in Arctic (60–90° N) annual mean net TOA radiative fluxes ($\Delta R_{\mathrm{TOA}}$) between conjugate warming and cooling patch experiments. **(b)** Differences in Arctic annual mean net surface radiative fluxes ($\Delta R_{\mathrm{sfc}}$). **(c)** Differences in heat fluxes due to the atmospheric heat transport to the Arctic regions ($\Delta R_{\mathrm{AHT}}$). **(d, e, f)** Responses of $\Delta R_{\mathrm{TOA}}$, $\Delta R_{\mathrm{sfc}}$, and $\Delta R_{\mathrm{AHT}}$ in the Antarctic regions. The location of each point denotes the center of the corresponding patch, and the colors denote the differences between conjugate warming and cooling patch experiments. Black circles indicate that the differences are statistically significant at a 95 % confidence level.

TOA radiation (Fig. 1a and d) and surface radiation (Fig. 1b and e) reflects atmospheric heat transport (Fig. 1c and f).

Figure 1a–c show the responses of the Arctic energy budgets to SST warmings in global oceanic regions. In response to western and central tropical Pacific SST warming, there is a significant increase in poleward energy transport toward the Arctic regions (Fig. 1c), as indicated by the positive poleward heat transport to the Arctic region (positive $\Delta R_{\mathrm{AHT}}$). This enhanced energy transport warms the Arctic atmosphere, leading to an increase in surface radiation (positive $\Delta R_{\mathrm{sfc}}$, Fig. 1b) due to higher surface and air temperatures. Simultaneously, the warmer atmosphere emits more longwave radiation into space, resulting in a decrease in TOA radiation (negative $\Delta R_{\mathrm{TOA}}$, Fig. 1a). Conversely, warming in the tropical Indian Ocean reduces the poleward energy transport to the Arctic region (negative $\Delta R_{\mathrm{AHT}}$), leading to cooler Arctic atmospheric temperatures, and there is a decrease in surface radiation (negative $\Delta R_{\mathrm{sfc}}$, Fig. 1b) and an increase in TOA radiation (positive $\Delta R_{\mathrm{TOA}}$, Fig. 1a). Sea surface warming in the mid-latitudes of the Northern Hemisphere increases Arctic surface radiation but has an insignificant impact on TOA radiation.

For the Antarctic energy budget, warming in the tropical Pacific and Indian oceans generally leads to increased poleward energy transport (positive $\Delta R_{\mathrm{AHT}}$, Fig. 1f), which warms the Antarctic atmosphere and results in increased Antarctic surface radiation (positive $\Delta R_{\mathrm{sfc}}$, Fig. 1e) and decreased Antarctic TOA radiation (negative $\Delta R_{\mathrm{TOA}}$, Fig. 1d). However, the response of $\Delta R_{\mathrm{TOA}}$ to warmings in the trop-ical Atlantic is positive (Fig. 1d). Warming in the Southern Ocean also leads to an increase in Antarctic surface radiation and a decrease in Antarctic TOA radiation. The Antarctic energy budget is generally not sensitive to warmings in subtropical regions. Both $\Delta R_{\mathrm{TOA}}$ and $\Delta R_{\mathrm{sfc}}$ decrease in response to warmings in patches centered at 60° S, because patches centered at 60° S cover part of the Antarctic region (60 to 90° S in this study), and the surface emits more energy into space as the sea surface warms, leading to a cooling radiative effect.

The response of Arctic $\Delta R_{\mathrm{AHT}}$ to tropical warmings is generally greater than Antarctic $\Delta R_{\mathrm{AHT}}$, indicating that more heat is transported to the Arctic region than to the Antarctic region when the tropics warm. This difference may partly contribute to faster Arctic warming than Antarctic warming under global warming.

The responses of the polar energy budget depend on the season. The spatial distribution of the Arctic polar energy budget response to regional SST in boreal winter (DJF) (Fig. 2a–c) is similar to the annual average values (Fig. 1a–c), but the magnitude of the DJF responses is greater than the annual responses. For the Antarctic region, the response of the DJF polar energy budget to tropical ocean warming aligns with the annual average values.

During boreal summer (JJA), the spatial distribution of the Arctic polar energy budget responses (Fig. 3a–c) differs from the annual average. Specifically, the responses of $\Delta R_{\mathrm{AHT}}$ to warmings in both the Indian Ocean and the western Pacific Ocean are positive. The responses of $\Delta R_{\mathrm{sfc}}$ and $\Delta R_{\mathrm{TOA}}$ in most patch experiments are statistically insignificant. Con-

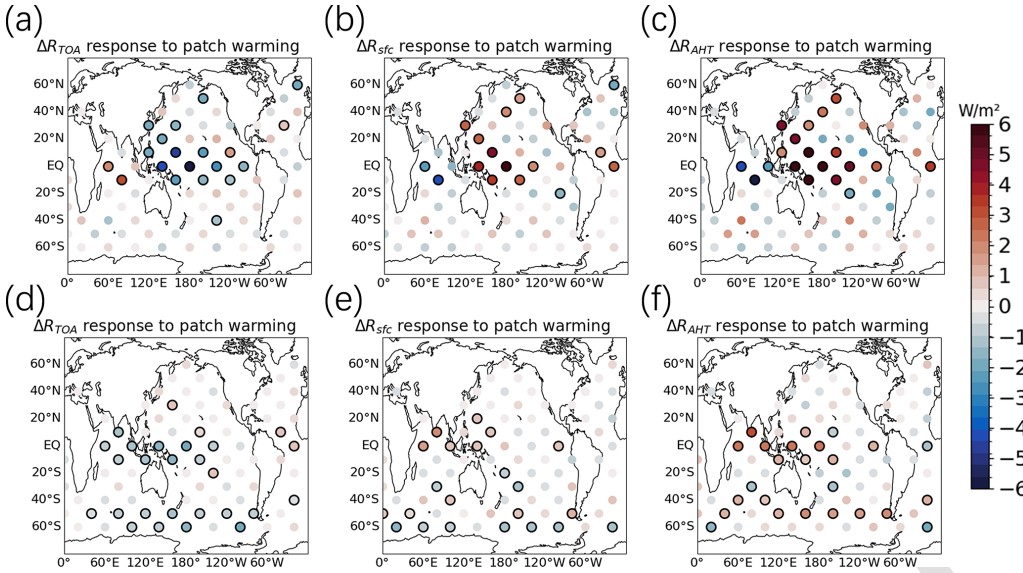

**Figure 2.** The same as Fig. 1, but for the DJF season.

versely, the Antarctic polar energy budget responses to tropical ocean warming in JJA are similar to the annual mean values.

## 3.2 Reconstruction using Green's function

The response of the polar energy budget to regional SST changes might be used to qualitatively explain how SST variations affect the polar energy budget.

The sensitivity of the polar energy budget to SST perturbations within a specific grid box, identified by index $i$, can be estimated using the following equation (Zhou et al., 2017):

$$\left(\frac{\partial R}{\partial \text{SST}_i}\right)_p = \frac{\sum_P \Delta \text{SST}_P \frac{\partial R}{\partial \text{SST}_P} \frac{S_i}{S_P}}{\sum_P \Delta \text{SST}_P}, \tag{7}$$

where $R$ denotes a specific energy flux; $\text{SST}_i$ denotes the SST in the $i$th grid box; $S_i$ and $S_P$ represent the ocean surface area of the specific grid point and the patch, respectively; and $\Delta \text{SST}_P$ is the SST anomaly of the $i$th grid in the $p$th patch. $\frac{\partial R}{\partial \text{SST}_P}$ is the sensitivity of $R$ to patch-averaged SST change, which is derived from experiments for the corresponding warming and cooling patch experiments. The sensitivity to grid boxes within a single patch corresponds to the average $R$ change due to a 1 K warming within that specific grid box. The sensitivities of the polar energy budget to SST perturbation in each grid box are shown in Fig. 4.

Utilizing these sensitivities, we can reconstruct the polar energy budget response to arbitrary changes in SST through Green's function approach, represented as

$$\Delta R = \Sigma_i \frac{\partial R}{\partial \text{SST}_i} \Delta \text{SST}_i + \varepsilon_i, \tag{8}$$

where $\varepsilon_i$ is an error term which represents the contributions from nonlinearities and non-SST-induced factors.

Then we use Green's function approach to reconstruct the $\Delta R_{\text{AHT}}$ in response to eight different SST patterns in the EOF-SST experiments (Fig. 5a–h). $\Delta R_{\text{AHT}}$ reconstructed by Green's function is then compared to model-produced values in the EOF-SST experiments (Fig. 5i and j). The results show that the majority of the experimental simulations of $\Delta R_{\text{AHT}}$ align closely with the $\Delta R_{\text{AHT}}$ reconstructed by Green's function, lying near the $y = x$ line. The biases of the values reconstructed by Green's function are partially induced by the SST change in the Arctic region, which is not captured by the Green's function reconstruction, and nonlinear terms also contribute to the bias. Therefore, Green's function approach can qualitatively explain how the SST perturbation patterns in Fig. 5a–h affect the polar energy budget.

## 3.3 Decomposition of polar energy budget responses with radiative kernels

To understand the mechanism of how the polar energy budget is affected by remote SST, we quantify the contributions of meteorological factors to the polar energy budget responses (Figs. 6 and 7) using radiative kernels (Huang et al., 2017).

Figure 6a shows that the contribution of cloud changes is relatively small compared to the Arctic $\Delta R_{\text{TOA}}$ response. Pacific SST warming results in a negative Arctic $\Delta R_{\text{TOA,cld}}$, whereas Indian Ocean warming generates a positive Arctic $\Delta R_{\text{TOA,cld}}$. Albedo exhibits a positive response in both the tropical Indian Ocean and the Pacific Ocean (Fig. 6a). The impact of Arctic $\Delta R_{\text{TOA,plk}}$ exhibits a positive response to SST increases in the tropical Indian Ocean and a negative response to the tropical western Pacific (Fig. 6c), indicating that the Planck feedback is the primary contributor to Arctic $\Delta R_{\text{TOA}}$. The sensitivity of Arctic $\Delta R_{\text{TOA,LR}}$ to SST in-

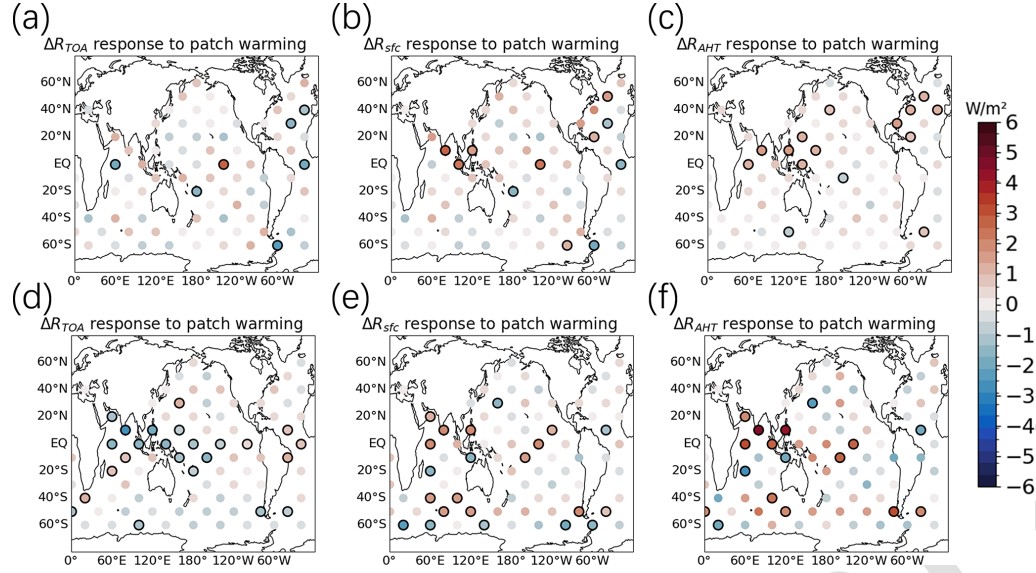

**Figure 3.** The same as Fig. 1, but for the JJA season.

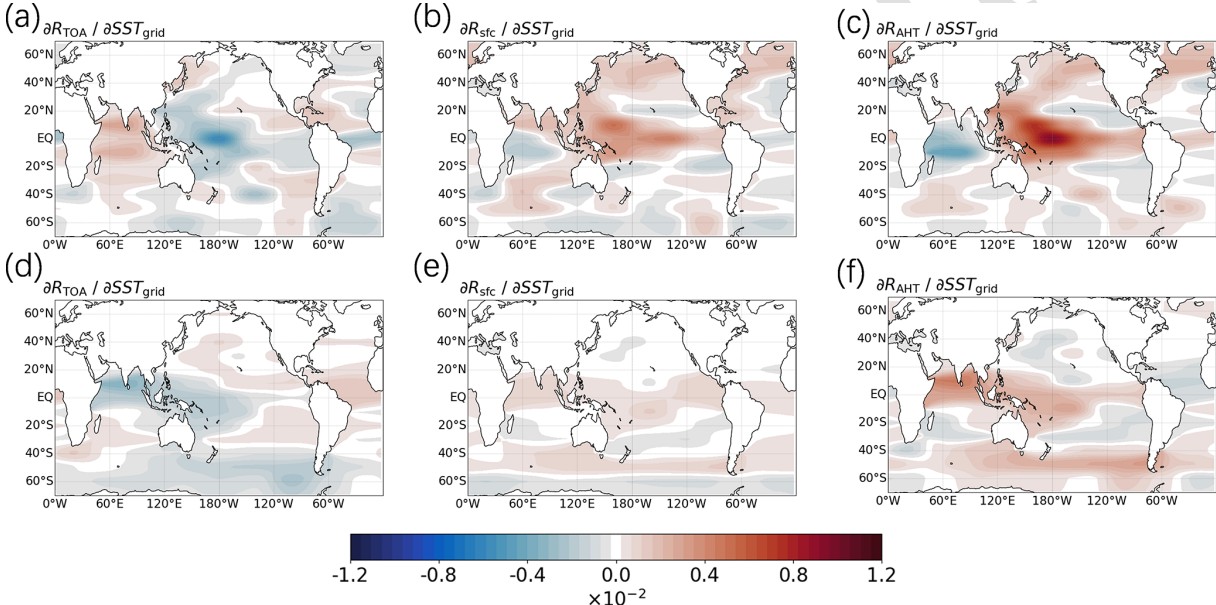

**Figure 4.** Sensitivity of **(a)** $\partial R_{\mathrm{TOA}}/\partial \mathrm{SST}_i$ of Arctic, **(b)** $\partial R_{\mathrm{sfc}}/\partial \mathrm{SST}_i$ of Arctic, **(c)** $\partial R_{\mathrm{AHT}}/\partial \mathrm{SST}_i$ of Arctic, **(d)** $\partial R_{\mathrm{TOA}}/\partial \mathrm{SST}_i$ of Antarctic, **(e)** $\partial R_{\mathrm{sfc}}/\partial \mathrm{SST}_i$ of Antarctic, and **(f)** $\partial R_{\mathrm{AHT}}/\partial \mathrm{SST}_i$ of Antarctic to surface warming in each grid box, calculated using Eq. (7). The units are $\mathrm{W\,m^{-2}\,K^{-1}}$.

creases exhibits a negative response to warming in the tropical western Pacific, while showing a positive response in the Indian Ocean (Fig. 6d). The Planck feedback's response to SST warming is approximately twice that of the lapse-rate feedback.

For the Antarctic, the contribution of clouds to the $\Delta R_{\mathrm{TOA}}$ response is minimal (Fig. 6f), as is the contribution of albedo changes to the Antarctic $\Delta R_{\mathrm{TOA}}$ response (Fig. 6e). In contrast to the Arctic, the Antarctic $\Delta R_{\mathrm{TOA}}$ response to SST

changes is jointly dominated by Planck and lapse-rate feedbacks. Warming in the Indian Ocean induces a stronger negative $\Delta R_{\mathrm{TOA,plk}}$ (Fig. 6g), whereas warming in the Pacific leads to a more pronounced negative $\Delta R_{\mathrm{TOA,LR}}$ (Fig. 6h). Additionally, warming in the tropical Atlantic triggers a notably strong positive $\Delta R_{\mathrm{TOA,LR}}$ (Fig. 6h).

Figure 7 shows the contributions of meteorological factors to the responses of Arctic and Antarctic $\Delta R_{\mathrm{sfc}}$. For the Arctic, the spatial distribution of clouds' contribution to Arc-

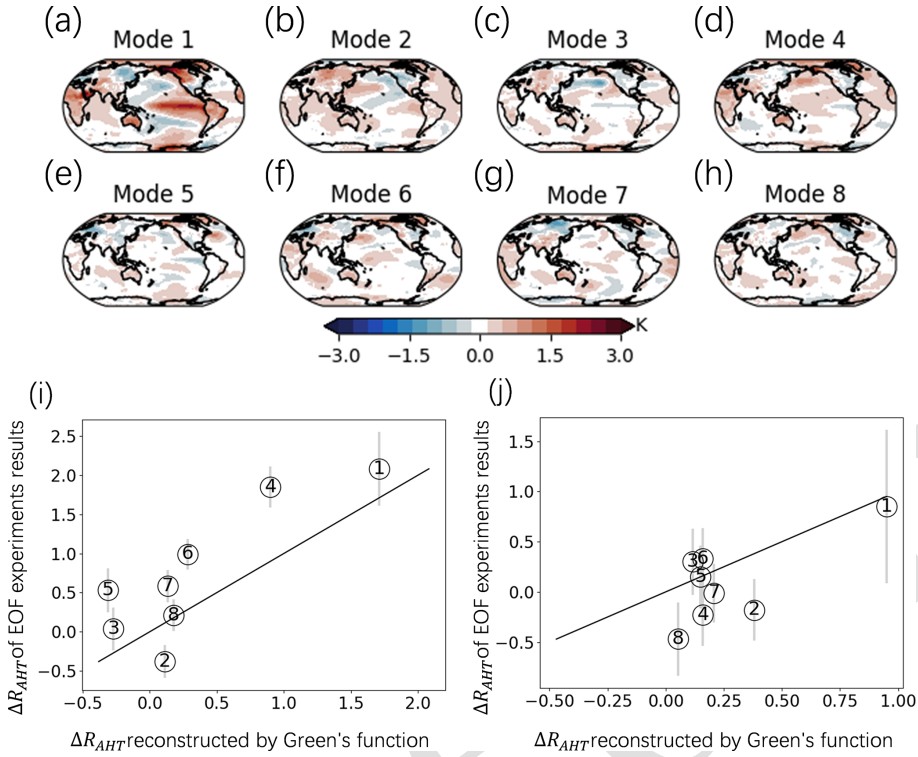

**Figure 5. (a–h)** Surface temperature change patterns in individual EOF-SST experiments. **(i)** Comparison of Arctic $\Delta R_{\mathrm{AHT}}$ responses to different SST change patterns in EOF-SST experiments (*y* axis) and that reconstructed by Green's function approach (*x* axis). All values are averaged annually in this figure. The digits represent the number of corresponding EOF modes in each experiment. Error bars correspond to the 95 % confidence interval. **(j)** Response of Antarctic $\Delta R_{\mathrm{AHT}}$.

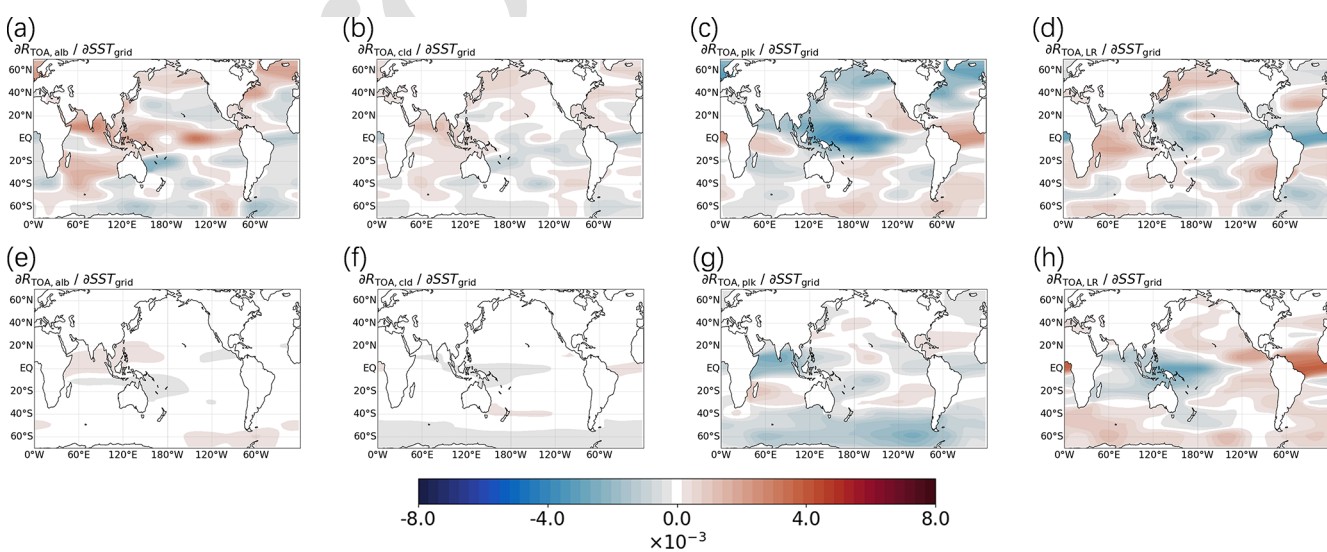

**Figure 6.** Sensitivity of annual mean $\Delta R_{\mathrm{TOA,cld}}$ **(a)**, $\Delta R_{\mathrm{TOA,alb}}$ **(b)**, $\Delta R_{\mathrm{TOA,plk}}$ **(c)**, $\Delta R_{\mathrm{TOA,LR}}$ **(d)** for Arctic and annual mean $\Delta R_{\mathrm{TOA,cld}}$ **(e)**, $\Delta R_{\mathrm{TOA,alb}}$ **(f)**, $\Delta R_{\mathrm{TOA,plk}}$ **(g)**, and $\Delta R_{\mathrm{TOA,LR}}$ **(h)** for Antarctica between conjugate warming and cooling patch experiments. The units are $\mathrm{W\,m^{-2}\,K^{-1}}$.

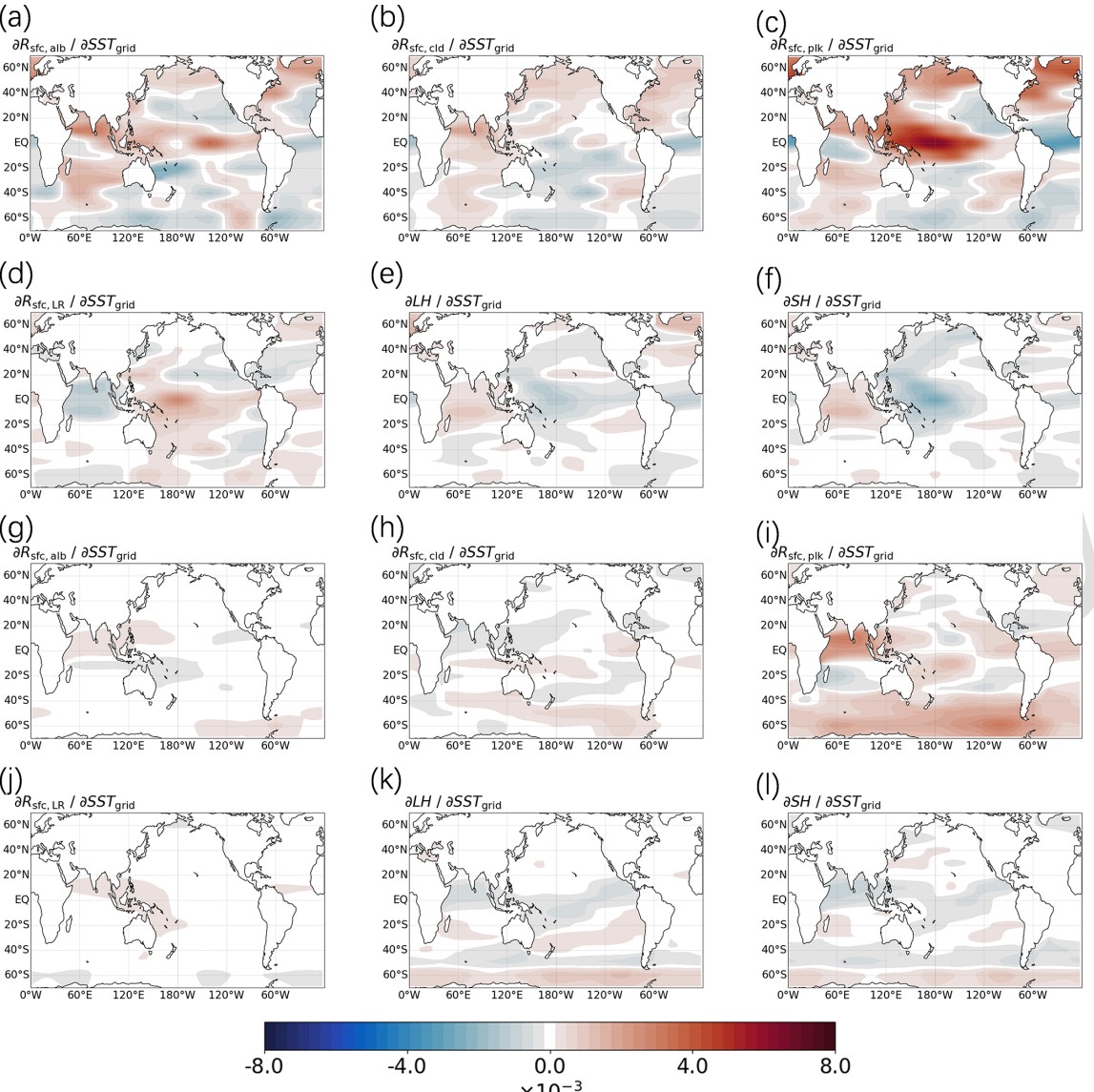

**Figure 7.** Sensitivity of annual mean $\Delta R_{\mathrm{sfc,cld}}$ (a), $\Delta R_{\mathrm{sfc,alb}}$ (b), $\Delta R_{\mathrm{sfc,plk}}$ (c), $\Delta R_{\mathrm{sfc,LR}}$ (g), $\Delta$LH (h), $\Delta$SH (i) for the Arctic region, and annual mean $\Delta R_{\mathrm{sfc,cld}}$ (d), $\Delta R_{\mathrm{sfc,alb}}$ (e), $\Delta R_{\mathrm{sfc,plk}}$ (f), $\Delta R_{\mathrm{sfc,LR}}$ (j), $\Delta$LH (k), and $\Delta$SH (l) for the Antarctic region between conjugate warming and cooling patch experiments. The units are $\mathrm{W\,m^{-2}\,K^{-1}}$.

tic $\Delta R_{\mathrm{sfc}}$ is similar to that for Arctic $\Delta R_{\mathrm{TOA}}$, but the values are significantly higher (Fig. 7b). The response of Arctic $\Delta R_{\mathrm{sfc,alb}}$ closely resembles that of $\Delta R_{\mathrm{TOA,alb}}$, with similar magnitudes (Fig. 7a). Similar to the response of Arctic $\Delta R_{\mathrm{TOA,plk}}$, the Arctic $\Delta R_{\mathrm{sfc,plk}}$ response is also strong (Fig. 7c), but the signs of $\Delta R_{\mathrm{sfc,plk}}$ responses are generally opposite to those of the $\Delta R_{\mathrm{TOA,plk}}$ response. The response of Arctic $\Delta R_{\mathrm{sfc,LR}}$ to tropical ocean warming also shows signs opposite to those of $\Delta R_{\mathrm{TOA,LR}}$ (Fig. 7d). Over the tropical Indian Ocean, Planck feedback is positive in the northern region and negative in the southern region, whereas lapse-rate feedback is consistently negative across the entire tropical Indian Ocean. Overall, Planck feedback remains dominant

for Arctic $\Delta R_{\mathrm{sfc}}$. The responses of Arctic $\Delta$LH and Arctic $\Delta$SH are negative in response to warmings in the tropical western Pacific and positive in response to warmings in the tropical Indian Ocean (Fig. 7e and f), although these factors have a minor impact on Arctic $\Delta R_{\mathrm{sfc}}$. The negative responses of Arctic $\Delta$LH and Arctic $\Delta$SH in response to warmings in the tropical western Pacific indicate a suppression of upward turbulent heat fluxes at the Arctic surface, primarily due to enhanced energy transport from lower latitudes into the Arctic region. As warm air masses are advected poleward, the associated increase in downward longwave radiation warms the Arctic surface. This warming stabilizes the lower atmospheric boundary layer, thereby reducing the vertical turbu-

lence necessary for effective heat exchange between the surface and the atmosphere.

Similar to the Arctic regions, the contribution of clouds to Antarctic $\Delta R_{sfc}$ is small (Fig. 7h). Albedo has no significant impact on Antarctic $\Delta R_{sfc}$ (Fig. 7g), because sea ice concentration is fixed in these patch experiments, and snow cover in Antarctica does not change significantly in these experiments. The Planck feedback dominates the contributions to Antarctic $\Delta R_{sfc}$, with Antarctic $\Delta R_{sfc,plk}$ responding positively to tropical ocean warming (Fig. 7i). Similar to the Arctic, the lapse-rate feedback contributes less significantly to $\Delta R_{sfc}$ (Fig. 7j). Antarctic $\Delta LH$ and Antarctic $\Delta SH$ show minimal responses, with opposite signs to $\Delta R_{sfc,plk}$ (Fig. 7k and l). These results suggest that while temperature adjustments are notable in Antarctica, the responses of cloud cover and surface heat fluxes to remote SST warming have a small impact on Antarctic $\Delta R_{sfc}$.

## 3.4 AHT responses to regional SST changes

According to Figs. 6 and 7, the Planck feedback, which is primarily driven by changes in temperature, is the primary contributor to the polar energy budget responses in these experiments. The sign of $\Delta R_{AHT}$ is generally the same as $\Delta R_{sfc,plk}$ and the opposite of $\Delta R_{TOA,plk}$. In addition, the difference between TOA and the surface energy budget reflects the contribution of changes in polar AHT. Therefore, AHT plays a critical role in determining the polar energy budget by changing the air temperature of polar regions. The responses of AHT to SST perturbations in the mid-latitudes are consistent with our intuition, but the opposite Arctic AHT responses to SST warming over the tropical Indian Ocean (TIO) and tropical Pacific Ocean (TPO) require further investigations.

To explore the underlying mechanisms of this phenomenon, we compare the climate responses to warmings in two illustrative patches within the TPO and TIO. The center of the illustrative TPO patch is $0°$ N, $180°$ E, and the center of the illustrative TIO is $0°$ N, $60°$ E.

Figure 8 presents the responses of surface temperature ($\Delta T_s$), 200 hPa geopotential height ($\Delta Z_{200}$), and 500 hPa geopotential height ($\Delta Z_{500}$) to warmings in these patches, providing background information to later AHT studies. Consistent with previous studies (Annamalai et al., 2007; Barsugli and Sardeshmukh, 2002; Ding et al., 2014), our experiments reveal that SST anomalies in different ocean basins induce contrasting atmospheric circulation patterns, primarily through Rossby wave responses affecting the Pacific–North American (PNA) pattern.

Specifically, warming in the TPO region leads to an increase in $\Delta T_s$ over the Tibetan Plateau, eastern Europe, tropical Africa, northeastern North America, and most of the Antarctic regions. Concurrently, $\Delta Z_{200}$ exhibits a local increase over the TPO region, a decrease over the North Pacific, and an increase over northeastern North America and Antarctic regions. The $\Delta Z_{500}$ response mirrors the $\Delta Z_{200}$

pattern but with reduced intensity. In contrast, warming in the TIO region induces $\Delta T_s$ increases over Antarctic regions and significant warming over the Indian subcontinent, while the Tibetan Plateau experiences cooling. Notably, the northwest of North America shows marked cooling under TIO warming. The $\Delta Z_{200}$ response to TIO warming displays a dipole pattern, characterized by increases in the tropical warming regions and decreases toward the poles, followed by subsequent increases. The $\Delta Z_{500}$ response follows a similar trend to $\Delta Z_{200}$, with weaker intensity.

To quantify the impacts of SST warming over the TPO and TIO on the Arctic AHT, we computed the AHT responses to the warming of the two patches separately. AHT can be calculated as the vertically integrated and zonally averaged transport of moist static energy ($S$). According to Neelin and Held (1987), $S$ can be defined as follows:

$$S = c_p T_a + L Q + g Z, \tag{9}$$

where $T_a$ represents atmospheric temperature, $c_p$ is the specific heat capacity of air at constant pressure, $L$ denotes the latent heat of vaporization of water, $Q$ is specific humidity, $g$ is the acceleration due to gravity, and $Z$ represents geopotential height. The components of $S$ will be denoted below as $S_T$, $S_Q$, and $S_Z$, respectively.

The poleward transport of moist static energy $S$ can be decomposed into mean meridional circulation (MOC), stationary eddy (SE), transient eddy (TE), and transient overturning circulation (TOC) components, following the methodologies of Priestley (1948) and Lorenz (1955). According to Donohoe et al. (2020), for each latitude $\theta$, atmospheric energy transport is

$$AHT(\theta) =$$

$$2\pi a \cos(\theta) \int_0^{P_s} [\overline{V}][\overline{S}] + \overline{[V^* S^*]} + \overline{[V'^* S'^*]} + \overline{[V]'[S]'} \frac{dp}{g}, \tag{10}$$

where $V$ represents the meridional velocity. Square brackets ([ ]) denote zonal averages, overbars denote time averages over each month of analysis, asterisks (*) are departures from the zonal average, and primes (′) are departures from the time average. The first term signifies the MOC driven by the vertical gradient in $S$, taking into account mass conservation in MOC energy transport. The second term is SE, showing poleward transport in warm or moist sectors. These first two terms can be calculated from monthly mean data. The third term pertains to the transport associated with TE, primarily baroclinic synoptic eddies. The fourth term involves energy transport by the covariance between zonal-mean overturning circulation and vertical stratification, referred to as TOC, which is significantly smaller than the other components and thus is often excluded in AHT discussions.

Figure 9 shows the changes in AHT and its components in response to warmings in the TPO and TIO. AHT response to warmings in the TPO at $60°$ N is positive, and AHT response to warmings in the TIO at $60°$ N is negative. For both cases,

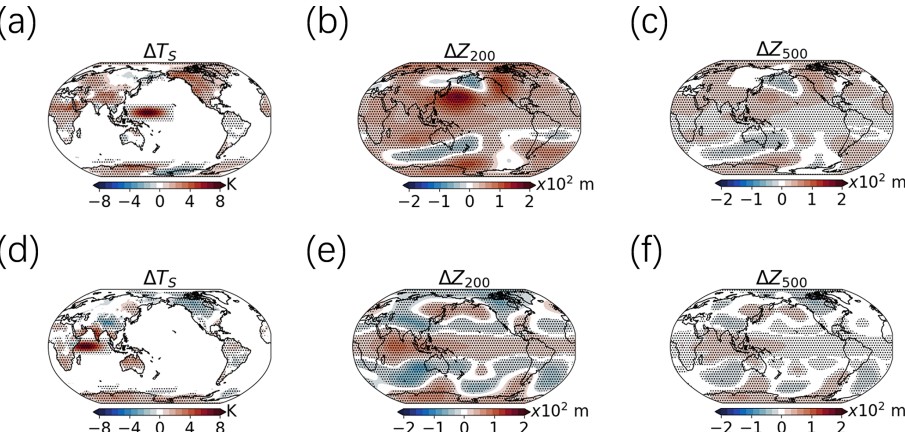

**Figure 8.** Spatial distributions of responses in $\Delta T_s$ (**a**), $\Delta Z_{200}$ (**b**), and $\Delta Z_{500}$ (**c**) following increased SST in a patch over the tropical Pacific Ocean (TPO) and in $\Delta T_s$ (**d**), $\Delta Z_{200}$ (**e**), and $\Delta Z_{500}$ (**f**) following increased SST in a patch over the tropical Indian Ocean (TIO). Dotted areas indicate regions that passed the 95 % confidence test.

AHT to the Arctic region is dominated by SE (Fig. 9b), and the opposite SE response to the TPO and TIO leads to opposite responses in AHT, which also causes different responses of TOA and surface energy budgets in the Arctic regions. Additionally, Fig. 9c and d further dissect the SE responses to warmings in the TPO and TIO, respectively. The results indicate that dry static energy predominantly drives the SE response to warmings in both the TPO and the TIO.

To better understand the opposite response of SE heat flux to warmings in the TPO and TIO, we analyzed the spatial distribution of SE heat fluxes. The vertically integrated SE heat flux ($\phi$) can be computed through the following integral:

$$\phi = \int_0^{P_s} V^* S^* \frac{\mathrm{d}p}{g} . \tag{11}$$

In response to warmings in the TPO, the vertically integrated SE heat flux exhibits significant oscillatory characteristics over the Pacific Ocean. According to Fig. 10a, $\phi$ increases in the tropical western and northwestern Pacific, decreases over the central Pacific, and then increases again over the northeastern Pacific and Alaska. Over land, $\phi$ increases over northeastern Asia, the Tibetan Plateau, and Europe. This phenomenon is consistent with Goss et al. (2016), who found that warming in the low-latitude Pacific leads to increased $\phi$ in higher latitudes.

Combining Eqs. (9) and (11), we are able to further attribute $\phi$ to changes in dry static heat flux ($\Delta\phi_{Ta}$), latent heat flux ($\Delta\phi_Q$), and potential energy ($\Delta\phi_Z$), respectively. Among the three main contributing factors to $\Delta\phi$, $\Delta\phi_{Ta}$ aligns closely with the overall $\Delta\phi$ pattern, indicating it as the primary contributor (Fig. 10b). The spatial pattern of $\Delta\phi_{Ta}$ can be explained by the change in $\Delta V^*$ and $\Delta S_{Ta}^*$ (Fig. 10c and d), which reflects the spatial pattern of stationary waves. In addition, $\Delta\phi_Q$ is large in the low latitudes but is small near the poles (Fig. 10e). Interestingly, $\Delta\phi_Z$ exhibits an opposite pattern to $\Delta\phi$ in the Northern Hemisphere (Fig. 10f). Despite

the changes in $\Delta S_Z^*$ that have a similar spatial distribution to $\Delta S_{Ta}^*$ (Fig. 10h), $\Delta\phi_Z$ shows an opposite trend to $\Delta\phi_{Ta}$ due to the differing correlation between $\Delta V^*$ and $\Delta S_Z^*$. This indicates that while $\Delta S_Z^*$ increases in regions where $\Delta V^*$ also increases, their combined effect on $\Delta\phi_Z$ leads to a negative contribution compared to $\Delta\phi$.

In response to warmings in the TIO, the spatial pattern of $\Delta\phi$ oscillation is quite different from the TPO case. The transfer of SE heat flux encounters obstacles near the Tibetan Plateau, which might explain why the response of $\Delta\phi$ is different in the Northern Hemisphere and the Southern Hemisphere. Globally, $\Delta\phi_{Ta}$ remains the dominant contributor to $\Delta\phi$, while $\Delta\phi_Q$'s contribution remains relatively low. Notably, in the Tibetan Plateau region, $\Delta\phi_Z$ becomes the primary driver of $\Delta\phi$ (Fig. 10i and n).

Based on these results, we are able to understand why the responses of AHT near 60° N to warmings in the TPO and TIO are different. The response of stationary waves to warmings in these two regions is different, so the poleward moist static energy transported by SE is also different, leading to opposite AHT responses. These findings support the research results of Goss et al. (2016) and the tropical excitation mechanism for Arctic warming outlined by Lee et al. (2011).

## 4   Conclusion

This study delves into the mechanisms behind the responses of the radiative budget in high-latitude regions to sea surface warmings in the low latitudes through a series of idealized SST change experiments. It elucidates the mechanisms through which the polar energy budget responds to distant SST variations, revealing significant different impacts of SST changes on the Arctic and Antarctic energy budgets across different oceanic regions. These impacts are mediated by alterations in AHT, the distribution of sensible heat flux from

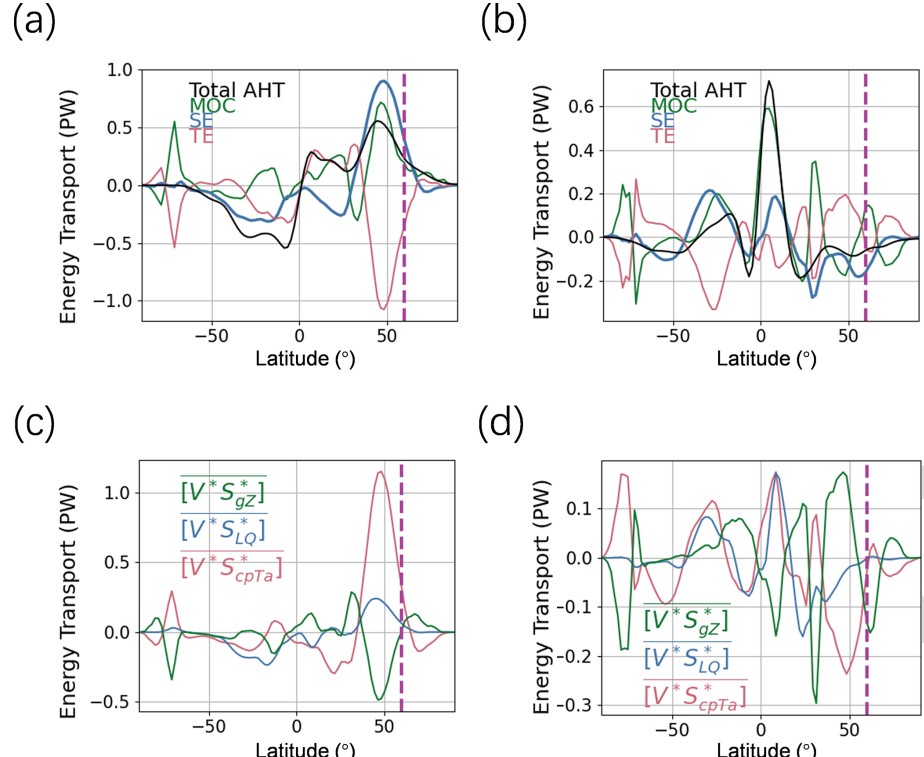

**Figure 9.** Decomposition of meridional AHT responses. Panels **(a)** and **(b)** show the changes in meridional AHT following SST warming in the TPO and TIO, respectively. The total AHT is represented by a thick black line, while the contributions from the MOC, SE, and TE are depicted by fine lines in green, blue, and red, respectively. Panels **(c)** and **(d)** detail the decomposition of the SE component from **(a)** and **(b)**, with the contributions from $Z$, $Q$, and $T$ shown in green, blue, and red, respectively. The dotted purple line represents the 60° N latitude line.

stationary eddies, and the interactions among various climatic drivers such as temperature, humidity, and cloud radiative processes.

Specifically, increases in SST in the Pacific and Indian oceans have opposite effects on the Arctic polar energy budget in boreal winter, attributable to different responses of stationary waves to warming in these oceans, which subsequently alter the patterns of poleward AHT. Warming of the Pacific SST tends to enhance heat transport to the Arctic, leading to Arctic air temperature increases, whereas warming in the Indian Ocean reduces the heat transport toward the Arctic, resulting in Arctic air temperature decreases. Additionally, the study highlights the fact that the response of the polar energy budget varies with the season. During boreal winter, the sensitivity of the Arctic polar energy budget to SST changes in tropical regions is stronger, indicating a higher sensitivity of the polar region to tropical ocean warming in winter. Using radiative kernels, the contributions of meteorological factors to the TOA radiation response were quantified. The results indicate that changes in Planck feedback are the primary contributor to changes in polar TOA radiation, while the contributions from clouds and albedo are relatively small. The decomposition of surface radiation also shows that the Planck feedback plays a primary role in driving changes in polar surface radiation. Finally, the study reconstructed the AHT responses under different EOF-SST modes using Green's function approach, validating the consistency between the model experiment results and the Green's function reconstructions. Although biases exist in certain EOF modes, partially due to SST changes within the polar regions and nonlinear effects, Green's function method generally provides a reasonable reconstruction of the polar energy budget response to SST changes.

The primary findings of the study are summarized as follows:

1. In response to SST warmings in most tropical and mid-latitude regions, polar air temperatures increase due to enhanced AHT, leading to an increase in the polar surface energy budget and a decrease in the polar TOA energy budget.

2. The response of Arctic AHT to warmings in the tropical Indian Ocean is negative in boreal winter. Stationary eddies play a crucial role in modulating the polar AHT response to tropical SST changes.

3. Subtropical SST changes have relatively weak impacts on the polar energy budget.

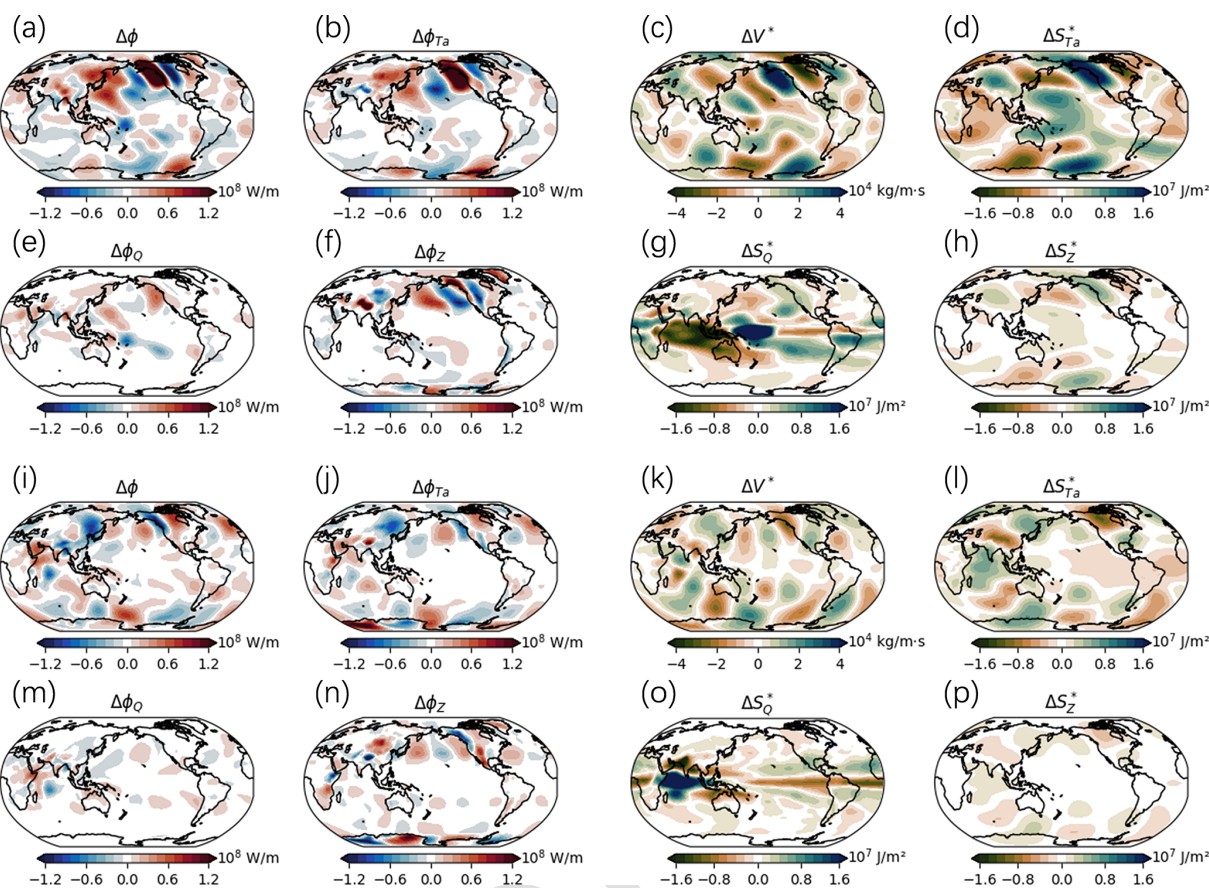

**Figure 10. (a)** $\Delta\phi$ induced by SST warming in the TPO. **(b)** The contribution of $\Delta\phi_{\mathrm{Ta}}$ to the SE. **(c)** The vertical integration of $\Delta V^*$ from the surface to the TOA. **(d)** The vertical integration of $\Delta S_{\mathrm{Ta}}^*$ from the surface to the TOA. **(e)** The contribution of $\Delta\phi_Q$ to the SE. **(f)** The contribution of $\Delta\phi_Z$ to the SE. **(g)** The vertical integration of $\Delta S_Q^*$ from the surface to the TOA. **(h)** The vertical integration of $\Delta S_Z^*$ from the surface to the TOA. **(i)**–**(p)** are the same as **(a)**–**(h)** but for responses to warming in the TIO.

The findings of this study have implications for understanding and predicting polar climate responses to global warming. The distinct responses of the Arctic and Antarctic energy budgets to regional SST changes underscore the necessity of considering regional specificity when modeling and predicting climate change. The different impacts of SST changes in various oceanic regions on the polar energy budgets highlight the importance of incorporating regional specificity in climate models. Moreover, the study underscores the important role of AHT in modulating polar temperatures, emphasizing the critical role of radiative feedbacks in shaping the polar climate. Understanding the mechanisms of AHT and its interaction with stationary eddies can lead to improved predictions of polar climate responses to global SST changes. The analyses of radiative feedbacks, including the roles of temperature, humidity, and clouds, provide a comprehensive understanding of the factors contributing to polar amplification. Results from only one global climate model are analyzed in this study, and analyses with more climate models from Green's Function Model Intercomparison

Project (GFMIP; Bloch-Johnson et al., 2024) might be useful to reduce model biases in future studies.

**Code availability.** The code used in this study is available upon request from the corresponding author.

**Data availability.** The model output data from the patch experiments are available at https://doi.org/10.5281/zenodo.8026626, https://doi.org/10.5281/zenodo.8025843, and https://doi.org/10.5281/zenodo.8169525 (Zhou, 2023a, b, c). The radiative kernels used in this study are based on ERA-Interim climatology and were generated using the RRTMG radiation scheme (Huang et al., 2017). These kernels are available from https://doi.org/10.17632/vmg3s67568.4 (Huang and Huang, 2023). The HadISST sea surface temperature dataset (Rayner et al., 2003) used in this study is publicly available from the Met Office Hadley Centre at https://www.metoffice.gov.uk/hadobs/hadisst/ (Met Office, 2025).

**Author contributions.** Methodology: CZ. Investigation: QW. Writing (original draft preparation): QW. Writing (review and editing): CZ, YL, and LZ.

**Competing interests.** The contact author has declared that none of the authors has any competing interests.

**Disclaimer.** Publisher's note: Copernicus Publications remains neutral with regard to jurisdictional claims made in the text, published maps, institutional affiliations, or any other geographical representation in this paper. While Copernicus Publications makes every effort to include appropriate place names, the final responsibility lies with the authors.

**Acknowledgements.** The numerical simulations were done on the computing facilities in the High-Performance Computing Center of Nanjing University.

**Financial support.** This research has been supported by the National Natural Science Foundation of China (grant no. 42375038). Yincheng Liu was supported by the Program for Outstanding PhD Candidates of Nanjing University (grant no. 202401B05).

**Review statement.** This paper was edited by Yuan Wang and reviewed by two anonymous referees.

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
