# Peer review of "Responses of polar energy budget to regional sea surface temperature changes in extra-polar regions Qingmin Wang, Yincheng Liu, Lujun Zhang, and Chen Zhou"

_EGUsphere, 2024_

## Author Comment (AC1)

**Responses to Reviewer #1**

We thank the reviewer for the valuable comments. The manuscript has been modified according to the suggestions. Below are our specific responses to the reviewer's comments.

*The authors use sea-surface temperature (SST) warming patch experiments to quantify how the Arctic energy budget is impacted by remote warming. They find a central role for changes in atmospheric heat transport, primarily due to stationary eddies, for connecting tropical SST changes to changes in Arctic surface and top-of-atmosphere radiation. They also highlight the opposite response of Arctic radiation to warming in the Indian Ocean versus Western Pacific, as a result of differences in the stationary eddy response.*

*The premise of this paper is novel and interesting: while past studies have explored the remote impacts of SST-patch changes on atmospheric circulation and global warming, the polar warming response has been less explored, and there are many open questions about the role of atmospheric heat transport for polar warming. This paper will be of interest to research communities studying polar climate change, the pattern effect and climate sensitivity, and the teleconnections between tropical SSTs and atmospheric circulation. However, I recommend some changes to the analysis to more clearly and mechanistically interpret the results and situate them within the context of previous literature.*

**Response:**

Thanks for the valuable comments. We have revised the paper to address all the comments.

*Major Comments:*

*Rossby wave response to tropical SST forcing*

*1. There are some previous relevant papers that would be helpful to add when discussing the opposite response to Indian Ocean versus tropical Pacific warming. Annamalai et al (2007; https://doi.org/10.1175/JCLI4156.1) have a review of some of these in their introduction paragraph 4, with a focus on how SST anomalies from different ocean basins affect the Pacific-North American (PNA) pattern. Barsugli and Sardeshmukh (2002) use SST patch experiments to show that warm SST anomalies in the tropical Pacific produce positive PNA index values, while warm SST anomalies in the Indian Ocean produce negative PNA index values, both by triggering a Rossby wave response. Others like Ding et al. (2014; https://doi.org/10.1038/nature13260) have connected this atmospheric circulation response to changes in Arctic warming. It seems like your experiments are consistent with these results: the Indian Ocean and tropical Pacific generate opposite temperature responses in the Arctic by producing different Rossby wave responses and changes in stationary eddy heat transport.*

**Response:**

Thanks for the suggestions. We added these references when we discuss the opposite response to Indian Ocean versus tropical Pacific warming (L278-282):

"*Figure 8 presents the responses of surface temperature ($\Delta T_s$), 200hPa geopotential height ($\Delta Z_{200}$), and 500hPa geopotential height ($\Delta Z_{500}$) to warmings in these patches, providing background information to later AHT studies. Consistent with previous studies (Annamalai et al., 2007; Barsugli & Sardeshmukh, 2002; Ding et al., 2014), our experiments reveal that SST anomalies in different ocean basins induce contrasting atmospheric circulation patterns, primarily through Rossby wave responses affecting the Pacific-North American (PNA) pattern.*"

*2. As in the references above, to investigate this Rossby wave response, can the authors plot the 200-hPa geopotential height response to these two SST experiments? It would be helpful to more clearly illustrate this mechanism linking tropical SST perturbations to changes in Arctic temperature and radiation.*

**Response:**

We plotted 200-hpa and 500-hpa geopotential height (Fig. 8) response to investigate the Rossby wave response. Below are the specific changes made to the manuscript (L278-290):

*"Figure 8 presents the responses of surface temperature ($\Delta T_s$), 200hPa geopotential height ($\Delta Z_{200}$), and 500hPa geopotential height ($\Delta Z_{500}$) to warmings in these patches, providing background information to later AHT studies. Consistent with previous studies (Annamalai et al., 2007; Barsugli & Sardeshmukh, 2002; Ding et al., 2014), our experiments reveal that SST anomalies in different ocean basins induce contrasting atmospheric circulation patterns, primarily through Rossby wave responses affecting the Pacific-North American (PNA) pattern.*

*Specifically, warming in the TPO region leads to $\Delta T_s$ increase over the Tibetan Plateau, Eastern Europe, tropical Africa, northeastern North America, and most of Antarctica. Concurrently, $\Delta Z_{200}$ exhibits a local increase over the TPO region, a decrease over the North Pacific, and increases over north-eastern North America and Antarctica. The $\Delta Z_{500}$ response mirrors the $\Delta Z_{200}$ pattern but with reduced intensity. In contrast, warming in the TIO region induces $\Delta T_s$ increases over Antarctica and significant warming over the Indian subcontinent, while the Tibetan Plateau experiences cooling. Notably, the northwest of North America shows marked cooling under TIO warming. The $\Delta Z_{200}$ response to TIO warming displays a dipole pattern, characterized by increases in the tropical warming regions and decreases toward the poles, followed by subsequent increases. The $\Delta Z_{500}$ response follows a similar trend to $\Delta Z_{200}$ with weaker intensity."*

*3. I think Equation (7) is wrong: In Kaspi and Schneider (2013) Equation (3), the stationary eddy response is defined as Vbar\*Sbar – Vbar,bracket\*Sbar,bracket, but the authors here have written (V\*S)bar – Vbar,bracket\*Sbar,bracket, which is actually equal to the stationary plus transient eddy response. This will impact the results shown in Figure 8. Also, Figure 8 has 16 panels—consider whether all are necessary.*

**Response:**

Thanks for pointing out the issue. We deleted the equation, and now we are directly calculating $\overline{V^*S^*}$ from our model data,

$$V^* = V - [V]$$

$$S^* = S - [S]$$

and updated Figure 8 (now Figure10) accordingly (L329-331):

*"To better understand the opposite response of SE heat flux to warmings in TPO and TIO, we analyzed the spatial distribution of SE heat fluxes. The vertically integrated SE heat flux ($\phi$) can be computed through the following integral:*

$$\phi = \int_0^{P_s} V^* S^* \frac{dp}{g} \,, \tag{11}"$$

Figure 10 explains how the change of V, $S_{Ta}$, $S_Q$ and $S_Z$ contribute to $\phi$ ($\phi_{Ta}$, $\phi_Q$, $\phi_Z$), so there are 8 panels for each case, and as a result there are 16 panels.

*4. Figure 6 and L199-212: I didn't find this figure helpful, other than illustrating that the Pacific patch warms the Arctic while the Indian Ocean patch cools the Arctic (although I would suggest a smaller scale for the color bar in 6a,e to be able to see the Arctic response). How do zonal-mean V and Q changes in Figure 6c and 6d help us understand this response (given that the authors later show stationary eddies are key, the covariance of V and MSE anomalies from the zonal mean would be more relevant)—please add mechanistic interpretation or remove this.*

**Response:**

Thank you for your valuable feedback. We have revised the content of Figure 6 (now Figure 8). It now presents "The spatial distribution of $\Delta T_s$ (a), $\Delta Z_{200}$ (b), and $\Delta Z_{500}$ (c) in response to an increase in SST over the TPO, and the spatial distribution of $T_s$ (d), $\Delta Z_{200}$ (e), and $\Delta Z_{500}$ (f) following an increase in SST over the TIO." Additionally, we have conducted an analysis as detailed in Response for major comments 1&2.

*Mechanisms of Arctic warming response*

*1. Introduction: Where is this statement that 50-85% of Arctic warming is induced by non-local drivers coming from (L48)? Some of the papers cited here (e.g. Stuecker et al., 2018) actually show the opposite—that very little polar amplification results from non-polar forcing. Papers like Dai et al. (2019) also show that local feedbacks due to sea-ice loss are needed to produce strong polar amplification. Pithan and Mauritsen (2014), Goosse et al. (2018), and Hahn et al. (2021) show that the local lapse-rate and albedo feedbacks contribute most to Arctic warming, followed by changes in poleward moisture transport. A more nuanced summary is needed: past studies have suggested a dominant role for local processes in driving polar amplification, but have also suggested that poleward moisture transport is another important contributor, and would support Arctic amplification even in the absence of local sea-ice feedbacks (e.g. Alexeev 2005). Moreover, local and remote processes interact, so remote heat transport may further contribute by amplifying local feedbacks.*

**Response:**

Sorry for the inappropriate expression. To avoid misunderstanding, we deleted these numbers: "Therefore, remote processes play an important role in driving Arctic warming, and the remote forcings are further amplified by local feedback processes."

In the previous draft, the 50-85% numbers denote the ratio of non-local forcings to total forcings (feedbacks are not regarded as forcings), which came from the following sentences:

In Chung and Räisänen (2011), they wrote: "the remotely-induced warming contributes more to the total annual-mean Arctic warming in ECHAM5 (≈85%) than in CAM3 (≈60%)."

In Taylor et al. (2022,), they wrote: "Chung and Räisänen (2011) attribute 60–85% of Arctic warming to non-local drivers, Yoshimori et al. (2017) find 60–70%, Park et al. (2018) ~50%, Shaw and Tan (2018) ~60%,…"    Link to Taylor et al. (2022):

https://www.frontiersin.org/journals/earth-science/articles/10.3389/feart.2021.758361/full

We revised the introduction to provide a more nuanced summary of the roles of local and remote processes in Arctic amplification (L27-57):

"*The polar energy budget is highly sensitive to various local feedback mechanisms. One important mechanism is the ice-albedo feedback. Global warming reduces snow cover and sea ice cover in the polar regions, leading to more solar radiation being absorbed, which in turn accelerates climate warming and further decreases albedo (Dickinson et al., 1987; Hall, 2004; Boeke and Taylor, 2018; Duan et al., 2019; Dai et al., 2019). Additionally, temperature feedback is another significant contributor to AA (Pithan and Mauritsen, 2014; Laîné et al., 2016; Sejas and Cai, 2016). It involves the processes of radiative cooling and is characterized by the Planck and lapse-rate feedbacks. The Planck feedback, driven by the nonlinear relationship between blackbody radiation and temperature, provides negative feedback to TOA fluxes at all latitudes, especially in low latitudes (Pierrehumbert, 2010). The lapse-rate feedback is a significant driver of AA: in the Arctic regions, stable stratification and temperature inversions trap surface warming and reduce radiative cooling, thereby enhancing warming. In contrast, the tropics experience significant upper-atmosphere warming due to convection, which does not similarly trap heat (Pithan and Mauritsen, 2014). During climate warming, the transformation of ice clouds into water clouds increases cloud albedo, leading to negative feedback (Mitchell et al., 1989; Li and Le Treut, 1992). Simultaneously, the decrease in lower tropospheric stability increases Arctic cloud cover and optical thickness (Barton*

*et al., 2012; Solomon et al., 2014; Taylor et al., 2015; Yu et al., 2019), contributing to Arctic autumn and winter warming (Boeke and Taylor, 2018). These local feedbacks are considered primary contributors to Arctic amplification (Pithan and Mauritsen, 2014; Goosse et al., 2018; Hahn et al., 2021; Dai et al., 2019).*

*Polar climate is also affected by remote influences, whose interaction drives Arctic warming (Li et al., 2021). While some studies suggest that remote forcing plays a relatively minor role in Arctic amplification (Stuecker et al., 2018), other research highlights the significant impact of poleward heat and moisture transport from lower latitudes in enhancing Arctic warming, and AA exists even in the absence of local sea-ice feedbacks (Alexeev et al., 2005; Graversen and Burtu, 2016). Specifically, poleward atmospheric heat transport (AHT) and moisture transport are critical components that contribute substantially to the observed warming in the Arctic.*

*Under global warming, the AHT from low latitudes is more effective in reaching the polar regions compared to the equatorward transfer from high latitudes (Alexeev et al., 2005; Chung and Räisänen, 2011; Park et al., 2018; Shaw and Tan, 2018; Semmler et al., 2020), and multiple global climate model experiments have been conducted to measure the remote influence on Arctic warming (Alexeev et al., 2005; Chung and Räisänen, 2011; Yoshimori et al., 2017; Park et al., 2018; Shaw and Tan, 2018; Stuecker et al., 2018; Semmler et al., 2020). The transport of water vapor from mid-latitudes also plays an important role by enhancing the greenhouse effect prior to condensation and increasing cloudiness after condensation, which together warm the Arctic during winter (Graversen and Burtu, 2016). Graversen and Burtu (2016) showed that latent heat transport can lead to significantly more Arctic warming than dry static energy (DSE) transport, even when delivering an equivalent amount of energy. Therefore, remote processes play an important role in driving Arctic warming, and the remote forcings are further amplified by local feedback processes.*"

*2. To understand the polar feedback and atmospheric heat transport response, I would recommend dividing the TOA radiation response (and heat transport convergence) (in W/m2) by the Arctic near-surface temperature response (in K), as in Kay et al. (2012; https://doi.org/10.1175/JCLI-D-11-00622.1). This would better show how remote warming impacts Arctic feedbacks and heat transport convergence.*

**Response:**

We understand the reviewer's suggestion to quantify the feedback strength by dividing the $\Delta R_{TOA}$ and $\Delta R_{AHT}$ by the Arctic $\Delta T_s$, and below is an illustrative figure. However, the main purpose of this paper is to analyze how SST anomalies in the mid/low latitudes affect the energy budget in the polar regions, while the effects of polar sea ice change (which are important in determining polar feedback parameter) are not analyzed in this paper, so the polar feedback parameter is not analyzed in this paper. To maintain the consistency and simplicity of the figures and to ensure a clear presentation of the results, we are not presenting these results in this paper.

[Figure]

**Figure R2. Response of annual mean Arctic $\Delta R_{TOA}$ normalized by annual mean Arctic temperature response, $\lambda_{TOA} = \Delta R_{TOA}/\Delta T_s$.**

*I would also consider expanding the current feedback decomposition to include the water vapor feedback and to split the temperature feedback into a Planck and lapse-rate response, consistent with previous studies. Similarly to Figure 9, can the authors also show the sensitivity of the Arctic-averaged near-surface temperature to the local SST changes? I also find Figure 9 with the Green's function approach to be more informative than figures with the individual patch responses like Figure 1, so would suggest combining the patch experiments to create maps like Figure 9 for the feedback analysis, too.*

**Response:**

We have expanded the current feedback decomposition to split the temperature feedback (fixed-RH) into Planck and lapse-rate responses. We are still using the fixed-RH decomposition framework because the RH value does not change significantly.

Following the reviewer's suggestion, we are now using sensitivity maps to perform the feedback analysis.(Figs. 6-7). Given the negligible sensitivity of $\partial R_{RH}/\partial SST_i$, we have decided not to

include its discussion in the paper (Figure R3).

[Figure]

Figure R3 Difference of annual mean $\partial R_{TOA,RH}/\partial SST_i$(a), $\partial R_{sfc,RH}/\partial SST_i$(b) for Arctic and annual mean $\partial R_{TOA,RH}/\partial SST_i$ (c), $\partial R_{sfc,RH}/\partial SST_i$ (d) for Antarctica between conjugate warming and cooling patch experiments.

***Minor Comments***

*L18: Suggest adding a sentence to the abstract to indicate why the reader should care about these results—what's the key takeaway, and what are the implications.*

**Response:**

We have added the following sentence to the abstract (L19-20):

 "*These results help explain how the polar climate is affected by the magnitude and spatial pattern of remote SST change.*"

*L23: "its lower albedo"—I don't think this is true, and would delete. Also would add a reference for Southern Ocean heat uptake in L24 (like Armour et al. 2016) alongside the elevation/feedback references that are here already (Salzmann and Hahn).*

**Response:** Yes, it should be "weaker albedo reduction" instead of "lower albedo". We changed the statement, and added a reference to Armour et al. (2016) to support the discussion on the Southern Ocean's heat uptake alongside the existing references (L24-26):

"*However, the mechanisms in Antarctica differ from the Arctic due to factors like the high elevation of the Antarctic ice sheet, weaker albedo reduction and strong Southern Ocean heat uptake, which delay the response (Salzmann, 2017; Armour et al., 2016; Hahn et al., 2021; Smith et al., 2019).*"

*L27: after "snow cover" add something like "and melts sea ice" (a huge contributor to the albedo feedback)*

**Response:** We have revised the sentence to include the melting of sea ice as a significant contributor to the albedo feedback (L28):

"*Global warming reduces snow cover and sea ice cover in the polar regions*".

*L30-33: Suggest editing this incomplete description of the temperature feedback's contribution to AA. The main mechanism in the cited Pithan and Mauritsen reference is the lapse-rate feedback—in which surface warming is trapped by surface temperature inversions and contributes little to warming at higher altitudes (unlike in the tropics), which leads to less efficient radiative cooling in the Arctic than in the tropics. The Planck feedback also contributes to AA—in part because surface warming starting from colder temperatures in the Arctic produces less outgoing longwave radiation than when starting from warmer temperatures in the tropics, following the Stefan-Boltzmann equation.*

**Response**:

We have revised this section, and a detailed response is provided in the reply to the major comment under 'Mechanisms of Arctic warming response'.

*L41: The phrasing of "efficiency" is vague—I would reword this. Also, the main point of the cited Stuecker et al. (2018) paper is the opposite of the point of this paragraph—they find that polar amplification is dominated by local, not remote forcing.*

**Response:**

We have revised this section, and a detailed response is provided in the reply to the major comment under 'Mechanisms of Arctic warming response'.

We reworded "efficiency" as "more effective"(L48):

"*Under global warming, the AHT from low latitudes is more effective ...*"

*L52: Consider just writing out "polar energy budget"—I don't think PEB is very common as an acronym, and it would be easier to read without the acronym.*

**Response:**

We have revised the manuscript by replacing all instances of "PEB" with "polar energy budget" to enhance readability.

*L59: Suggest rewording this sentence, as the literature supports a large role of synoptic-scale waves for poleward heat transport—synoptic-scale transient eddies contribute significantly to both mean-state poleward heat transport and its changes under increased CO2 (e.g., Donohoe et al., 2020: https://doi.org/10.1175/JCLI-D-19-0797.1). I think the authors are saying that planetary waves are more important for the response to tropical warming, but should make this clearer.*

**Response:**

We revised this sentence following the comment (L61-67):

"*For instance, intensified convective activity within the Pacific Warm Pool not only strengthens the propagation of Rossby waves toward the poles but also increases the frequency of these fluctuations. This enhancement in Rossby wave activity boosts the transport of water vapor to the Arctic, augmenting the downward longwave radiation in the Arctic regions (Rodgers et al., 2003; Lee et al., 2011; Lee, 2012; Lee, 2014). While synoptic-scale transient eddies contribute significantly to mean-state poleward heat transport and its changes under increased CO2 (Donohoe et al., 2020), their overall impact is relatively minor compared to that of amplified planetary waves in responses*

*to tropical warming (Baggett and Lee et al., 2017).*"

*L90: What magnitude of SST anomaly, A, is imposed?*

**Response:** We added information on it (L95-96):

"*… which is set to be +4 K and -4 K in this study*".

*L111: Should say "western and central tropical Pacific," not eastern?*

**Response:**

We have corrected the text in L144 from "eastern tropical Pacific" to "western and central tropical Pacific".

*L110-120: This comes across as a descriptive list rather than telling a cohesive and interesting story. The authors might instead consider first discussing the advective, TOA, and surface responses to the tropical Pacific warming, and then the advective, TOA, and surface responses to the Indian Ocean warming. It would be helpful to add some mechanistic interpretation here, too, like these results suggest that in response to tropical Pacific warming, there is increased poleward atmospheric heat transport, which warms the Arctic atmosphere and therefore increases TOA radiative cooling and surface radiative heating. Also considering the rest of the paper, it would be generally helpful to include more mechanistic interpretation.*

**Response:** We have revised the paper accordingly, optimizing the logical order of the content to ensure a clearer structure and a more coherent narrative (L143-160):

"*Figures 1(a-c) show the responses of the Arctic energy budgets to SST warmings in global oceanic regions. In response to western and central tropical Pacific SST warming, there is a significant increase in poleward energy transport towards the Arctic regions (Figure 1c), as indicated by the positive poleward heat transport to the Arctic region (positive $\Delta R_{AHT}$). This enhanced energy transport warms the Arctic atmosphere, leading to an increase in surface radiation (positive $\Delta R_{sfc}$, Figure 1b) due to higher surface and air temperatures. Simultaneously, the warmer atmosphere emits more longwave radiation to space, resulting in a decrease in TOA radiation (negative $\Delta R_{TOA}$, Figure 1a). Conversely, warming in the tropical Indian Ocean reduces the poleward energy transport to the Arctic region (negative $\Delta R_{AHT}$), leading to cooler Arctic atmospheric temperatures, and there is a decrease in surface radiation (negative $\Delta R_{sfc}$, Figure 1b) and increase in TOA radiation (positive $\Delta R_{TOA}$, Figure 1a). Sea surface warming in the midlatitudes of the northern*

*hemisphere increases Arctic surface radiation, but has insignificant impact on TOA radiation.*

*For the Antarctic energy budget, warming in the tropical Pacific and Indian Oceans generally leads to increased poleward energy transport (positive $\Delta R_{AHT}$, Figure 1f), which warms the Antarctic atmosphere and results in increased Antarctic surface radiation (positive $\Delta R_{sfc}$, Figure 1e) and decreased Antarctic TOA radiation (negative $\Delta R_{TOA}$, Figure 1d). However, the response of $\Delta R_{TOA}$ to warmings in the tropical Atlantic is positive (Figure 1d). Warming in the Southern Ocean also leads to an increase of Antarctic surface radiation and decrease in Antarctic TOA radiation. Antarctic energy budget is generally not sensitive to warmings in subtropical regions. Both $\Delta R_{TOA}$ and $\Delta R_{sfc}$ decrease in response to warmings in patches centred at 60°S, because patches centred at 60°S cover part of the Antarctic region (60°S to 90°S in this study), and the surface emit more energy to space as the sea surface warms, leading to a cooling radiative effect."*

*L252: Should be Kaspi and Schneider (2013). Many of the other citations in the text are also missing "et al."—suggest checking the citation formatting throughout the paper.*

**Response:**

We have deleted the citation "Kaspi and Schneider (2013)." Additionally, we have reviewed all citations throughout the manuscript and have updated them to ensure proper formatting, including the use of "et al."

---

## Author Comment (AC2)

**Responses to Reviewer #2**

We thank the reviewer for the valuable comments. The manuscript has been modified according to the suggestions. Below are our specific responses to the reviewer's comments.

*The authors investigated the influence of the regional SST change on polar amplification through a set of idealized SST patch experiments. Their findings indicate that sea surface warming in most tropical regions enhances poleward energy transport, with the exception of the Indian Ocean, which is due to different responses of stationary waves. The innovative method employed is commendable, and the results are reasonable. I would recommend a minor revision for this paper.*

*Arctic is experiencing a faster warming rate than the global average during the recent decade, and the underlying reasons for this amplification remain somewhat unclear. Previous energy budget analyses, such as Pithan & Mauritsen (2014), showed the significance of local feedbacks. Stucker et al. (2018) demonstrated through model simulations that Arctic amplification is primarily driven by local forcing and feedbacks. However, Ding et al. (2017) highlighted the role of circulation in influencing September sea-ice extent. Some observations also indicate short-period warming events in the Arctic often follow a period of anomalous energy transport. This highlights the necessity for a deeper understanding of how changes in poleward energy transport interact with local feedbacks in the Arctic region. This paper makes a valuable contribution to addressing this important question.*

**Response:**

Thanks for the valuable comments. We have revised the paper to address all the comments.

*Major Comments:*

*1. The motivation of this paper could be further clarified in the Introduction section. The authors provide a substantial summary of the ongoing debate regarding the drivers of Arctic amplification (AA), specifically whether it is driven by local processes or remote factors. While it seems the authors will discuss the importance of atmospheric heat transport (AHT) later on, this point is not revisited in detail. The authors summarized that 50%-85% of Arctic warming is induced by non-local drivers in the Introduction, which also seems overstated. Given that this paper specifically focuses on how SST warming patterns influence the Arctic rather than quantifying the relative contributions from local and nonlocal drivers, I recommend that the authors either include a discussion or quantification of how their results support the importance of AHT in AA, or step back to enhance the literature review in the Introduction regarding the influence of SST warming patterns on AA. This would help readers understand what has been explored and what this paper aims to contribute.*

**Response:**

Sorry for the inappropriate expression. To avoid misunderstanding, we deleted these numbers: "Therefore, remote processes play an important role in driving Arctic warming, and the remote forcings are further amplified by local feedback processes."

In the previous draft, the 50-85% numbers denote the ratio of non-local forcings to total forcings (feedbacks are not regarded as forcings), which came from the following sentences:

In Chung and Räisänen (2011), they wrote: "the remotely-induced warming contributes more to the total annual-mean Arctic warming in ECHAM5 (≈85%) than in CAM3 (≈60%)."

In Taylor et al. (2022,), they wrote: "Chung and Räisänen (2011) attribute 60–85% of Arctic warming to non-local drivers, Yoshimori et al. (2017) find 60–70%, Park et al. (2018) ~50%, Shaw and Tan (2018) ~60%,…"    Link to Taylor et al. (2022):

  https://www.frontiersin.org/journals/earth-science/articles/10.3389/feart.2021.758361/full

We revised the introduction to provide a more nuanced summary of the roles of local and remote processes in Arctic amplification (L42-57):

*"Polar climate is also affected by remote influences, whose interaction drives Arctic warming (Li et al., 2021). While some studies suggest that remote forcing plays a relatively minor role in Arctic amplification (Stuecker et al., 2018), other research highlights the significant impact of poleward*

*heat and moisture transport from lower latitudes in enhancing Arctic warming, and AA exists even in the absence of local sea-ice feedbacks (Alexeev et al., 2005; Graversen and Burtu, 2016). Specifically, poleward atmospheric heat transport (AHT) and moisture transport are critical components that contribute substantially to the observed warming in the Arctic.*

*Under global warming, the AHT from low latitudes is more effective in reaching the polar regions compared to the equatorward transfer from high latitudes (Alexeev et al., 2005; Chung and Räisänen, 2011; Park et al., 2018; Shaw and Tan, 2018; Semmler et al., 2020), and multiple global climate model experiments have been conducted to measure the remote influence on Arctic warming (Alexeev et al., 2005; Chung and Räisänen, 2011; Yoshimori et al., 2017; Park et al., 2018; Shaw and Tan, 2018; Stuecker et al., 2018; Semmler et al., 2020). The transport of water vapor from mid-latitudes also plays an important role by enhancing the greenhouse effect prior to condensation and increasing cloudiness after condensation, which together warm the Arctic during winter (Graversen and Burtu, 2016). Graversen and Burtu (2016) showed that latent heat transport can lead to significantly more Arctic warming than dry static energy (DSE) transport, even when delivering an equivalent amount of energy. Therefore, remote processes play an important role in driving Arctic warming, and the remote forcings are further amplified by local feedback processes.*"

*2. The slower warming rate of the Antarctic is another interesting question. Since the authors have quantified how SST patches influence the energy budget in both the Arctic and Antarctic, I wonder if it could be possible to further discuss how the SST warming patterns might influence the asymmetry of AHT in polar regions. This may provide additional insights into the contrasting warming rates observed in the two areas.*

**Response:**

Yes, we agree with this point. We added a paragraph to discuss it (L161-163):

"*The response of Arctic $\Delta R_{AHT}$ to tropical warmings is generally greater than Antarctic $\Delta R_{AHT}$, indicating that more heat is transported to the Arctic region than that to the Antarctic region when the tropics warms. This difference may partly contribute to the faster Arctic warming than Antarctic warming under global warming.*"

**Minor Comments:**

*L8: "The results show…". This sentence is quite general; it would be beneficial to provide more specific details.*

**Response:**

Thanks for your feedback. We added a sentence beginning with "specifically" to explain this sentence (L11-13):

"*Specifically, an increase of poleward atmospheric energy transport to polar regions results in an increase of surface and air temperature, and the corresponding Planck feedback leads to a radiative warming at surface and radiative cooling at TOA.*"

*L23: "which is also applicable to the Antarctic". The mechanisms of Antarctic warming are different from the Arctic. The mechanism studies regarding the two regions are always separate.*

**Response:**

We've revised it (L24-26):

"*However, the mechanisms in the Antarctic region differ from the Arctic region due to factors like the high elevation of the Antarctic ice sheet, weaker albedo reduction and strong Southern Ocean heat uptake, which delay the response (Salzmann, 2017; Armour et al., 2016; Hahn et al., 2021; Smith et al., 2019).*"

*L48: "50%-85% of Arctic warming". This number is followed by several cited papers. This number is much higher than expected. I recommend the authors clarify the scenarios and methods used to obtain the number to avoid confusion.*

**Response:**

We have revised this section, see the reply to the major comment.

*L68: "(Lee, 2011; 2012; 1204". Typo, the closing parenthesis is missing.*

**Response:**

We have corrected the missing closing parenthesis

*L111: "The response of …". This is an important conclusion, but this sentence is difficult to understand. Suggest rephrasing.*

**Response:**

Thank you for your suggestion. We have reorganized our description of Figure 1 to improve clarity and make the important conclusion easier to understand (L143-160):

"*Figures 1(a-c) show the responses of the Arctic energy budgets to SST warmings in global oceanic regions. In response to western and central tropical Pacific SST warming, there is a significant increase in poleward energy transport towards the Arctic regions (Figure 1c), as indicated by the positive poleward heat transport to the Arctic region (positive $\Delta R_{AHT}$). This enhanced energy transport warms the Arctic atmosphere, leading to an increase in surface radiation (positive $\Delta R_{sfc}$, Figure 1b) due to higher surface and air temperatures. Simultaneously, the warmer atmosphere emits more longwave radiation to space, resulting in a decrease in TOA radiation (negative $\Delta R_{TOA}$, Figure 1a). Conversely, warming in the tropical Indian Ocean reduces the poleward energy transport to the Arctic region (negative $\Delta R_{AHT}$), leading to cooler Arctic atmospheric temperatures, and there is a decrease in surface radiation (negative $\Delta R_{sfc}$, Figure 1b) and increase in TOA radiation (positive $\Delta R_{TOA}$, Figure 1a). Sea surface warming in the midlatitudes of the northern hemisphere increases Arctic surface radiation, but has insignificant impact on TOA radiation.*

*For the Antarctic energy budget, warming in the tropical Pacific and Indian Oceans generally leads to increased poleward energy transport (positive $\Delta R_{AHT}$, Figure 1f), which warms the Antarctic atmosphere and results in increased Antarctic surface radiation (positive $\Delta R_{sfc}$, Figure 1e) and decreased Antarctic TOA radiation (negative $\Delta R_{TOA}$, Figure 1d). However, the response of $\Delta R_{TOA}$ to warmings in the tropical Atlantic is positive (Figure 1d). Warming in the Southern Ocean also leads to an increase of Antarctic surface radiation and decrease in Antarctic TOA radiation. Antarctic energy budget is generally not sensitive to warmings in subtropical regions. Both $\Delta R_{TOA}$ and $\Delta R_{sfc}$ decrease in response to warmings in patches centred at 60°S, because patches centred at 60°S cover part of the Antarctic region (60°S to 90°S in this study), and the surface emit more energy to space as the sea surface warms, leading to a cooling radiative effect.*"

L252: "Yohai". Typo, there's an extra space.

**Response:**

We've removed this reference.

*L357: "This knowledge…". This paper emphasizes the role of AHT, rather than the observed radiation in the Arctic.*

**Response:**

We have deleted this sentence.

*Figures 1-5: The colors are a bit faint, making it difficult to clearly distinguish the points.*

**Response:**

We have modified the color bars of Figures 1–5 (now Figures 1, 2, 3, and 6, 7), using smaller intervals.

*Figure 7: The black is not bolded, while the blue line is bolded.*

**Response:**

The different line weights were intentionally designed to highlight the role of SE, represented by the blue line. By making the blue line bold, we aim to emphasize its significance in the analysis.

---

## Author Comment (AC4)

**Responses to Editor #1**

We thank the editor for handling our submission and the reviewer for the valuable comments. The manuscript has been modified according to the suggestions. Below are our specific responses to the reviewer's comments.

*This study, through extensive simulation experiments and complex diagnostic analyses, explores the response of polar energy budget to sea surface temperature anomalies. The conclusion of this paper is clear. However, this paper still requires revision and further clarification.*

**Response:**

We thank the reviewer for the valuable comments.

*Major comments:*

*1.    The model description is unclear. I suppose the model used in this paper is CAM (only uncoupled atmosphere component), not CESM (coupled). The patch experiments are also incomplete. Does each warm or cold patch experiment consist of 80 sub-experiments, and every sub-experiments utilize different SST anomalies? What is the integration time of the sub-experiment?*

**Response:**

We thank the reviewer for their valuable feedback. Now we use "CESM1.2.1-CAM5.3" to describe the model. While CAM is the atmospheric module of CESM, other modules of CESM is also used when we perform the simulations. For example, CLM 4.0 (Community Land Model) and CICE (Community Ice Code) modules are active in our simulations.

*2.    In the method section, authors should provide a detailed introduction to the radiative kernels technology.*

**Response:**

We have provided a more detailed description and illustration of radiative kernel decomposition methodology in section 2.3 (L112-135):

"*2.3 Radiative Kernel Decomposition Methodology*

*This study employs the radiative kernel approach (Soden et al., 2008, Huang et al., 2017) to decompose both surface and TOA radiation into the radiative effects of various meteorological*

*variables, measured in watts per square meter (Wm⁻²). The core calculation involves multiplying the radiative kernels with the monthly anomalies of the corresponding climate fields as follows::*

$$\Delta R_X = K_X \cdot \Delta X \tag{5}$$

*where $X$ denotes an arbitrary non-cloud climate variable, $\Delta R_X$ represents the radiative effect at the surface or TOA associated with that variable, $K_X$ is the corresponding radiative kernel, and $\Delta X$ is the monthly anomaly of the climate variable, calculated as the deviation from the monthly climatological average. Positive values of $\Delta R$ indicate an increase in net incoming radiation, which corresponds to a warming effect on the Earth. The radiative kernels used in this analysis are derived from the ERA-Interim climatological fields and have been validated to perform well with climate model surface outputs (Huang et al., 2017; Liu et al., 2024).*

*Cloud radiative effects are calculated following the methodology of Soden et al. (2008):*

$$\Delta R_{cld} = \Delta CRF - \sum_X (K_X - K_X^0)\, \Delta X \tag{6}$$

*In this equation, $\Delta R_{cld}$ denotes the cloud-induced radiative anomalies, and CRF (Cloud Radiative Forcing) is defined as the difference in surface net radiation fluxes between all-sky and clear-sky conditions. The superscript $^0$ means the clear-sky kernels..*

*Building upon this framework, the study further decomposes the TOA and surface radiative anomalies into specific feedback components to achieve a more detailed analysis of the factors influencing the Earth's radiation balance. $\Delta R_{TOA}$ is partitioned into cloud-induced radiative anomalies ($\Delta R_{TOA,cld}$), albedo-induced radiative anomalies ($\Delta R_{TOA,alb}$), Planck feedback-induced radiative anomalies ($\Delta R_{TOA,plk}$), and lapse-rate feedback-induced radiative anomalies ($\Delta R_{TOA,LR}$). Similarly, $\Delta R_{sfc}$ is broken down into cloud-induced surface radiative anomalies ($\Delta R_{sfc,cld}$), albedo-related surface radiative anomalies ($\Delta R_{sfc,alb}$), Planck feedback-induced surface radiative anomalies ($\Delta R_{sfc,plk}$), lapse-rate feedback-induced surface radiative anomalies ($\Delta R_{sfc,LR}$), LH anomalies ($\Delta LH$) and SH anomalies ($\Delta SH$)."*

*3. Lines 48-49, "50%-85% of Arctic warming is induced by non-local drivers", this conclusion is a great shock and is certainly not a mainstream view. The references provided by the authors is also not compelling.*

Response:

Sorry for the inappropriate expression. To avoid misunderstanding, we deleted these numbers: "Therefore, remote processes play an important role in driving Arctic warming, and the remote

forcings are further amplified by local feedback processes."

In the previous draft, the 50-85% numbers denote the ratio of non-local forcings to total forcings (feedbacks are not regarded as forcings), which came from the following sentences:

In Chung and Räisänen (2011), they wrote: "the remotely-induced warming contributes more to the total annual-mean Arctic warming in ECHAM5 (≈85%) than in CAM3 (≈60%)."

In Taylor et al. (2022,), they wrote: "Chung and Räisänen (2011) attribute 60–85% of Arctic warming to non-local drivers, Yoshimori et al. (2017) find 60–70%, Park et al. (2018) ~50%, Shaw and Tan (2018) ~60%,…"    Link to Taylor et al. (2022):

https://www.frontiersin.org/journals/earth-science/articles/10.3389/feart.2021.758361/full

We revised the introduction to provide a more nuanced summary of the roles of local and remote processes in Arctic amplification (L42-57):

"*Polar climate is also affected by remote influences, whose interaction drives Arctic warming (Li et al., 2021). While some studies suggest that remote forcing plays a relatively minor role in Arctic amplification (Stuecker et al., 2018), other research highlights the significant impact of poleward heat and moisture transport from lower latitudes in enhancing Arctic warming, and AA exists even in the absence of local sea-ice feedbacks (Alexeev et al., 2005; Graversen and Burtu, 2016). Specifically, poleward atmospheric heat transport (AHT) and moisture transport are critical components that contribute substantially to the observed warming in the Arctic.*

*Under global warming, the AHT from low latitudes is more effective in reaching the polar regions compared to the equatorward transfer from high latitudes (Alexeev et al., 2005; Chung and Räisänen, 2011; Park et al., 2018; Shaw and Tan, 2018; Semmler et al., 2020), and multiple global climate model experiments have been conducted to measure the remote influence on Arctic warming (Alexeev et al., 2005; Chung and Räisänen, 2011; Yoshimori et al., 2017; Park et al., 2018; Shaw and Tan, 2018; Stuecker et al., 2018; Semmler et al., 2020). The transport of water vapor from mid-latitudes also plays an important role by enhancing the greenhouse effect prior to condensation and increasing cloudiness after condensation, which together warm the Arctic during winter (Graversen and Burtu, 2016). Graversen and Burtu (2016) showed that latent heat transport can lead to significantly more Arctic warming than dry static energy (DSE) transport, even when delivering an equivalent amount of energy. Therefore, remote processes play an important role in driving Arctic warming, and the remote forcings are further amplified by local feedback processes.*"

*4.  Lines 173-175, The responses of Arctic RLH and RSH are negative. This is very interesting, and I encourage the authors further discuss it in detail. I suppose that the warm Arctic caused by energy transport from low latitude suppresses the Arctic surface turbulent heat flux (increase the downward turbulent heat flux). Since the directions of LH and SH are positive upward (see minor comments #3), the responses of LH and SH are negative. It is worth noting that if the Arctic warming is driven by the local drivers, such sea ice reduction, it will lead to upward turbulent heat flux anomalies on the Arctic surface. The surface heat flux in the Arctic has shown an upward anomaly in recent years, and the warming of the Arctic should be dominated by local factors (see major comments #3).*

**Response:**

Thank you for your insightful comments and for highlighting the negative responses of Arctic latent heat flux and sensible heat flux in our study.

Yes, we agree that the negative responses indicate a suppression of upward turbulent heat fluxes at the Arctic surface. We interpret this suppression as primarily resulting from enhanced energy transport from lower latitudes into the Arctic region. Specifically, as warm air masses are advected poleward, the associated increase in downward longwave radiation warms the Arctic surface. This surface warming stabilizes the lower atmospheric boundary layer, thereby reducing the vertical turbulence necessary for effective heat exchange between the surface and the atmosphere. Consequently, both LH and SH, which represent the upward fluxes of latent and sensible heat respectively, decrease, leading to their negative anomalies.

Regarding local drivers like sea ice reduction leading to upward turbulent heat flux anomalies, we plan to analyze them with idealized sea ice experiments in the future. (It is likely that sea ice reduction leads to an increase of LH and SH, because the surface temperature increases in response to sea ice reduction)

We have discussed the negative response of LH and SH (L249-254):

"*The negative responses of Arctic ΔLH and ΔSH in response to warmings in the tropical western Pacific indicate a suppression of upward turbulent heat fluxes at the Arctic surface, primarily due to enhanced energy transport from lower latitudes into the Arctic region. As warm air masses are advected poleward, the associated increase in downward longwave radiation warms the Arctic*

*surface. This warming stabilizes the lower atmospheric boundary layer, thereby reducing the vertical turbulence necessary for effective heat exchange between the surface and the atmosphere."*

*5. Line 222, Should a barotropic mass-flux correction be applied before the computation of the energy transport? See the paper for more details.*

*Graversen RG. 2006. Do changes in midlatitude circulation have any impact on the Arctic surface air temperature trend?. J. Clim. 19: 5422–5438.*

**Response:**

Thank you for your valuable comments.

In this study, model simulation data are used to analyze energy transport. The model simulation data inherently incorporate mass-flux processes through their internal parameterization schemes, ensuring the physical consistency of atmospheric dynamics. Therefore, an additional barotropic mass-flux correction is not necessary for this study.

*Minor comments:*

*Lines 31-33, I cannot agree that the temperature feedback proposed by the authors creates a feedback loop.*

**Response:**

We thank the reviewer for the valuable comments. We are now using "Planck feedback" and "lapse rate feedback" to describe the feedback processes. These two terms are commonly used in climate feedback studies (L32-L38):

*"The Planck feedback, driven by the nonlinear relationship between blackbody radiation and temperature, provides negative feedback to TOA fluxes at all latitudes, especially in low latitudes (Pierrehumbert, 2010). The lapse-rate feedback is a significant driver of AA: in the Arctic regions, stable stratification and temperature inversions trap surface warming and reduce radiative cooling, thereby enhancing warming. In contrast, the tropics experience significant upper-atmosphere warming due to convection, which does not similarly trap heat (Pithan and Mauritsen, 2014). During climate warming, the transformation of ice clouds into water clouds increases cloud albedo, leading to negative feedback (Mitchell et al., 1989; Li and Le Treut, 1992)."*

*Line 80, remove the brackets.*

**Response:**

We've corrected it.

*Line 98. The directions of SH and LH are not defined, I assume they are positive upward.*

**Response:**

Yes, in our manuscript, SH and LH are defined as positive in the upward direction. The relevant clarification has been added (L105-106):

*"Additionally, both $SH$ and $LH$ are defined as positive upward."*

*Line 111, western tropical Pacific?*

**Response:**

Thank you for identifying the issue. We have revised the description of Figure 1 to reflect these changes appropriately (L143-145):

*"In response to western and central tropical Pacific SST warming, there is a significant increase in poleward energy transport towards the Arctic regions (Figure 1c),"*

*Line 112, western tropical Pacific?*

**Response:**

We have revised it.

*Lines 115-117, the descriptions of the Rsfc responses are not consistent with Figure 1b.*

**Response:**

We have completely revised our description of Figure 1.

*Line 119-120, the descriptions of the Radv responses are not consistent with Figure 1c.*

**Response:**

Thank you for your comments, as answered in the previous comment, we have completely revised our description of Figure 1.

*Lines 128-129, I can't understand how the Radv calculated at 60° What is the physical meaning? I suppose this should be the mean Radv north of 60°N.*

**Response:**

Yes, it is averaged from 60-90°N using Eq. (4), which reflects the heat transport acrossing 60°N. We changed the statement of $R_{adv}$ (now $R_{AHT}$): "heat fluxes resulting from atmospheric heat transport to the polar regions", and it is averaged as the mean value north of 60°N.

*Line 124, The negative response of Rsfc around 60°S may attributed to the Antarctic mean calculations of Rsfc (60°S-90°S). The imposed warm SST south of 60°S will increase the surface upward heat flux, thus the negative Rsfc.*

**Response:**

Thanks to your comments, we have included this idea in our revision of Figure 1 (L158-160):

"*Both $\Delta R_{TOA}$ and $\Delta R_{sfc}$ decrease in response to warmings in patches centred at 60°S, because patches centred at 60°S cover part of the Antarctic region (60°S to 90°S in this study), and the surface emit more energy to space as the sea surface warms, leading to a cooling radiative effect.*"

*Line 152, the tropical western Pacific.*

**Response:**

We've revised it.

*Lines 154-155, the contribution of cloud is important because it is statistically significant, despite the value is relatively small.*

**Response:**

Thank you for your valuable feedback regarding the contribution of clouds. We have updated the

manuscript (L223-224):

"*Figure 6a shows that the contribution of cloud changes is relatively small to the Arctic $\Delta R_{TOA}$ response. Pacific SST warming results in a negative Arctic $\Delta R_{TOA,cld}$, whereas Indian Ocean warming generates a positive Arctic $\Delta R_{TOA,cld}$.*"

Note that in response to another reviewer's suggestion, we re-plotted the cloud response, and the new figure does not include a statistical significance analysis.

*Line 158, western tropical pacific*

**Response:**

We've revised it.

*Line 159, eastern tropical pacific.*

**Response:**

We've revised it.

*Lines 191-192, Do the authors consider Radv to partly represent AHT?*

**Response:**

Thanks for the comment. We have replaced all instances of $\Delta R_{adv}$ with $\Delta R_{AHT}$.

*There is no significance test in Figure 6.*

**Response:**

We tested Figure 6 (now Figure 8) for significance, L292.

*Line 240, Figure 7a and Figure 7b.*

**Response:**

We've corrected it.

*Lines 276-277, I didn't noticed the poleward propagation of SE in the tropical warm pool.*

**Response:**

It should be the poleward moist static energy transported by SE. We have removed this sentence.

---

## Author Response (AR2)

Responses to Reviewer

We sincerely thank the reviewer for their constructive comments and thoughtful suggestions, as well as the editor for their time and dedicated efforts throughout the review process. Their input has been invaluable in helping us improve the quality and clarity of the manuscript.

*The authors have effectively addressed my concerns and suggestions, and the manuscript has been significantly improved. I particularly appreciate their thorough review of how energy transport influences Arctic amplification, which is important and highly relevant for linking this work to existing studies. As a result, the motivation for this work is now clearly articulated.*

*I have two minor suggestions for further improvement:*

*1. Line 24: More than half of the paragraph explains why warming in the Antarctic is slower than in the Arctic. While this is an interesting point, it doesn't seem directly relevant to the main topic of this paper. Since this is the opening paragraph, I suggest the authors reconsider this paragraph.*

**Response:**
We have shortened this sentence as suggested. Since the Antarctic energy budget is also analyzed in this study, we agree that a concise version is more appropriate. Please refer to Line 23 for the revised sentence:
"On the other hand, the Antarctic warming is weaker compared to the Arctic warming due to higher average elevation of the Antarctic continent, lower albedo, feedback efficiency differences, and the Southern Ocean's heat absorption capacity (Marshall 2003; Salzmann et al., 2017; Smith et al., 2019; Hahn et al., 2021)."

*2. Line 108: It would be helpful to add one or two sentences to explain why the first eight EOF modes were selected. Specifically, how much variance is explained by each mode? This will help readers focus on the predominated modes and their impact on AA.*

**Response:**
We have added a sentence clarifying this point: "The first eight EOF modes explain approximately 55% of the total variance in global SST anomalies, thereby representing the predominant variability patterns." Please see Line 106.